# Global variability of high nutrient low chlorophyll regions using neural networks and wavelet coherence analysis.

Gotzon Basterretxea[1], Joan S. Font-Muñoz[1], Ismael Hernández-Carrasco[2], Sergio A. Sañudo-Wilhelmy [3]

[1] Department of Marine Ecology, Instituto Mediterráneo de Estudios Avanzados, IMEDEA (UIB-CSIC), Miquel Marqués 21, 07190 Esporles, Illes Balears, Spain.
[2] Department of Oceanography and Global Change, Instituto Mediterráneo de Estudios Avanzados, IMEDEA (UIB-CSIC), Miquel Marqués 21, 07190 Esporles, Illes Balears, Spain.
[3] Department of Biological Sciences and Department of Earth Sciences, University of Southern California, Marine Biology and Biological Oceanography, Los Angeles, California 90089-0371, United States.

*Correspondence to*: Gotzon Basterretxea (gotzon@imedea.uib-csic.es)

**Abstract.** We examine 20 years of monthly global ocean color data and modeling outputs of nutrients using self-organizing map (SOM) analysis to identify characteristic spatial and temporal patterns of High Nutrient Low Chlorophyll (HNLC) regions and their association with different climate modes. The global nitrate to chlorophyll ratio threshold of NO3:Chl>17 (mmol NO3/mg Chl) is estimated to be a good indicator of the distribution limit of this unproductive biome that, on average, covers $92 \times 10^6$ km$^2$ (~25% of the ocean). The trends in satellite-derived surface chlorophyll ($0.6 \pm 0.4$ to $2 \pm 0.4\%$ yr$^{-1}$) suggest that HNLC regions in polar and subpolar areas have experienced an increase in phytoplankton biomass over the last decades but much of this variation is produced by a climate-driven transition in 2009-2010. Indeed, since 2010, the extent of the HNLC zones has decreased at the poles (up to 8%) and slightly increased at the equator (<0.5%). Our study finds that chlorophyll variations in HNLC regions respond to major climate variability signals Chlorophyll variations in HNLC regions respond to major climate variability signals such as El Niño Southern Oscillation (ENSO) and Meridional Overturning Circulation (MOC) at both short (2-4 years) and long (decadal) timescales. These results suggest global coupling in the functioning of distant biogeochemical regions.

## 1 Introduction

High nutrient low chlorophyll (HNLC) areas are ocean regions where primary production should be potentially high but phytoplankton biomass remains relatively low and constant despite the perennial nutrient availability for growth (Martin and Fitzwater, 1988; Chisholm and Morel, 1991). They are interesting regions because they challenge the accepted paradigm of a positive relation between macronutrient concentrations and phytoplankton biomass in open waters but, most importantly, because they represent an important fraction of the global ocean carbon budgets and, therefore, their extent influences the potential withdrawal of atmospheric $CO_2$ to the deep ocean (Martin et al., 1990; de Baar et

al., 1995; Boyd et al., 2005). It is estimated that HNLC biomes roughly cover between 20 and 30% of the world's oceans (Pitchford and Brintley, 1999; Tyrrel et al., 2005) comprising three major ocean areas; the Subarctic North Pacific (SNP), the Eastern Equatorial Pacific (EEP) and most of the Southern Ocean (SO) (Martin, 1990; Coale et al., 1996; Parekh et al., 2005).

Because nitrogen is the mineral nutrient needed in greatest abundance by phytoplankton and owing to its generalized depletion in surface waters over much of the oceans, it is considered a key limiting nutrient for ocean production. In HNLC regions, where nitrogen is in excess, other non-exclusive factors such as rapid top-down control by zooplankton grazing, low irradiance, limitations by silicic acid availability, and/or iron (Fe) limitation, have been hypothesized to explain the persistently low chlorophyll (Chl). While these factors may contribute in different degrees to the observed low Chl and determine the phytoplankton dynamics in HNLC regions (see Chavez et al., 1991; Cullen, 1995; Coale et al., 1996; Dugdale and Wilkerson, 1998; Landry et al., 2011), it is generally acknowledged that Fe availability is central to the productivity of HNLC regions (Boyd et al., 2007). All HNLC regions share a chronic Fe-depletion in surface waters and experimental results show highly positive productivity responses to Fe addition (Martin et al., 1994; Boyd et al., 2000, 2004; Tsuda et al., 2003; Coale et al., 2004). Indeed, iron is required in largest amounts than any of the trace metals for several metabolic processes, and not surprisingly, it has been considered the ultimate limiting nutrient (Moore and Doney, 2007). This has led to propose a conceptual model of phytoplankton nutrient limitation in the modern ocean based on two functioning regimes: one in which the supply of nutrients is relatively slow and nitrogen availability limits productivity, and a complementary regime, with enhanced nutrient supply, where Fe often limits productivity (Moore et al., 2013).

Iron limitation influences the uptake of nitrogen thereby explaining the unused nitrate concentrations in HNLC regions. Indeed, it has been proposed that a delicate balance between nitrogen and Fe availability modulates phytoplankton growth and that co-limitation is rather ubiquitous in the sea (Bryant, 2003; Browning et al., 2017). Other elements and compounds such as B-vitamins, which are also scarce in Fe-limited areas, can also be co-limiting factors for phytoplankton growth in these regions (e.g. Koch et al., 2011; Bertrand et al., 2012). For example, it has been experimentally shown that the addition of Fe and B12 to Antarctic phytoplankton assemblages can synergistically increase phytoplankton growth (Bertrand et al., 2011; Cohen et al., 2017).

Despite their relevance for global ocean productivity and carbon fluxes, HNLC regions remain loosely defined and knowledge of their temporal and spatial variability and trends is limited. Moreover, their response in a global warming scenario is uncertain. Only general aspects such as expected shifts in phytoplankton community composition or changes in Fe-cycling rates have been addressed to date (Fu et al., 2016; Lauderdale et al., 2020). The original description of HNLC systems by Minas et al. (1986)

referred to a slowly growing phytoplankton standing stock despite the presence of high nutrient concentrations. However, there are no rigid criteria accurately defining the functioning of these ecosystems. Several ecosystem characteristics such as species composition, biological structure, carbon utilization pathways, and response to climate change also differ between the HNLC and other ecosystems, reflecting differences in the limiting factor (e.g. Falkowski et al., 1998; Ono et al., 2008).

Of particular interest are the aspects related to the reduced variability and high permanence (i.e. temporal persistence) typically characterizing large HNLC regions. These features are distinctive from those of highly variable systems, which may temporarily present HNLC conditions. For example, some light-limited regions in high latitudes may present low productivity and enhanced nutrients during winter but it responds to a transient situation that does not correspond to the generally accepted HNLC paradigm. Similarly, high nutrients and low Chl have been observed at the end of the spring bloom in some productive systems (Nielsdóttir et al., 2009) and some areas located in coastal upwelling regions (Hutchins et al., 1998, 2002; Firme et al., 2003; Eldridge et al., 2004). While complying with the necessary conditions of high nutrient and low Chl, it is uncertain whether these ephemeral systems share structural and functioning similitudes with the large HNLC regions.

At a time when understanding biogeochemical responses to large-scale forcings, including climate change, has become a scientific priority, it seems appropriate to revisit some concepts of the functioning of HNLC regions. Their extent and variability are indicative of the dynamic changes in the bidirectional interrelationships of phytoplankton with the environment and with other organisms at large scales. Most of the information on the long-term variations of HNLC regions is depicted from global studies suggesting that their productivity is declining and that they experience prominent interannual to decadal fluctuations superimposed on these long-term trends (i.e. Boyce et al., 2010). Available evidence suggests that some HNLC regions may be decreasing in size as a result of increased ocean stratification (Ono et al., 2008). More recently, Yasunaka et al., (2016), determined that surface trends of phosphate and silicate in the North Pacific are associated with the shoaling of the mixed layer, reporting that surface nutrient concentration was correlated with the North Pacific Gyre Oscillation (NPGO). Some studies have shown that oligotrophic areas in the northern hemisphere are expanding between 0.8 and 4% per year, with a faster increase in winter months (Polovina et al., 2008). However, with some exceptions (e.g. Radenac et al., 2012; Yasunaka et al., 2014), specific long-term studies on HNLC regions are scarce and knowledge on their variability in the global ocean scale and their response to climate change remain uncertain.

The objective of the present study is to provide a quantitative assessment of the large-scale patterns of variability of the three major HNLC regions (SNP, EEP, and SO) and their relationship with the main modes of climate variability. Systematically determining the boundaries of these HNLC regions has remained elusive since it requires coherent information on nutrients and Chl fields. The present study is

based on the analysis of 20-year time series of monthly global ocean color data and nutrient concentrations
from a biogeochemical model using machine learning techniques and wavelets analysis. First, based on
the statistical analysis of global NO3:Chl ratios, we determine a robust quantitative criterion to objectively
define HNLC regions. Then we characterize the temporal variability patterns of HNLC regions based on
their NO3 and Chl concentrations by using the Self-Organizing Map (SOM) technique. We use the herein-
established statistical criterion to assess the spatial variations of HNLC regions over the study period
unveiled from the SOM analysis in the spatial domain of NO3:Chl ratios. Finally, through a combined
SOM-wavelet coherence analysis (WCA), we quantify the spectral power and the dynamic relationship
between the observed Chl variability and two main global-scale forcings; El Niño Southern Oscillation
(ENSO); and Meridional Overturning Circulation (MOC). We show that the combination of WCA with
SOM-derived characteristic time-series is an especially suitable tool for the analysis of driver-response
relationships in the ocean.
**2 Materials and Methods**

15       *2.1 Ocean color data*

We employ 20 years of monthly global composites of satellite Chl Level-3 products, derived from
merging SeaWiFS, MERIS, MODIS AQUA, and VIIRS sensors using a GSM algorithm (Maritorena and
Siegel, 2005), obtained from GlobColour data set (www.globcolour.info). The chlorophyll product is
spatially gridded, and the weighted average of the different merged Level-2 products is then calculated.
The composite consists of a rectangular regular map product in degrees with a spatial resolution of 0.25º
(i.e. around 28 km at the equator that varies with the latitude) and covers the period from January 1998
to December 2017. We excluded results in the Arctic Ocean and the coastal Southern Ocean due to the
interference of ice cover and prolonged gaps in the data. A total of 654395 pixels were considered in the
analysis. We are aware that the consistency of merged multi-mission ocean color satellite series may
suffer from some limitations influencing long trend analysis (Mèlin et al., 2017). However, no significant
increase or decrease is observed in the first-order trends of GlobColour data in more recent studies (e.g.
Moradi 2021). Therefore, while recognizing that some differences in regional and seasonal biases may
occur in unified data products and, acknowledging that discontinuities and trends of the median with time
should be interpreted carefully according to the sensors used (Garnesson et al., 2019), merged Chl can be
generally considered a good indicator of the magnitude of the overall phytoplankton trends.

*2.2 Nitrate data*

Since nutrient observations are still too scarce to allow obtaining time-resolved global-scale fields, we
used global NO3 obtained from the biogeochemical hindcast model provided by Mercator-ocean
(http://marine.copernicus.eu, see Fig. S1). Data on climate indices were obtained from available

databases. Bi-monthly Multivariate El Niño Southern Oscillation Index (MEI.v2), hereafter ENSO index, was obtained from the National Oceanic and Atmospheric Administration National Center for Environmental Prediction website (https://www.esrl.noaa.gov/psd/enso/mei/). MOC data (Smeed et al., 2019; Moat et al., 2022) for the period (2004-2018) was obtained from the RAPID-WATCH MOC monitoring project (www.rapid.ac.uk/rapidmoc).obtained using the PISCES model (Aumont et al.,, 2015). The model is forced by daily mean fields of ocean, sea ice, and atmospheric conditions. Ocean and sea ice forcings are obtained from the numerical simulation FREEGLORYS2V4 produced at Mercator-Ocean and the source of atmospheric forcings is the ERA-Interim reanalysis produced at ECMWF. Initial conditions are set from the World Ocean Atlas 2013 climatology. A complete model description can be found at (http://cmems-resources.cls.fr/documents/).

### 2.3 Climatological data

We compared available observational nutrient data (NO3) from the upper 20 m of the water column, obtained by merging bottle cast data from the World Ocean Database (WOD18, Boyer et al., 2018; https://www.nodc.noaa.gov/), with model results. Generally, we found good agreement between nitrate in situ data and model results ($r$=0.98). Main deviations occur in the Southern Ocean where NO3 concentrations are slightly overestimated (up to 7.2 mmol m$^{-3}$) and in some coastal areas affected by river runoff.

### 2.4 Identification of HNLC regions

Presently, the best approximation to define the global distribution of HNLC regions in the world ocean is the use of NODC maps of surface nutrients (https://www.nodc.noaa.gov/). However, excess nutrient availability by itself does not necessarily reflect HNLC conditions. In situ experiments are capable to discern Fe limitation conditions but a more manageable metric to assess the limits on the spatial extent of HNLC regions is required, in particular for remote sensing applications, as well as for allowing objective comparison between different environmental scenarios and studies.

To obtain a quantitative criterion for the definition of HNLC regions, we analyze the values of NO3:Chl ratios (mmol/mg) obtained from the SOM analysis on the time domain over the global ocean throughout the 20 years of data to identify a common statistical behavior representing HNLC conditions. In particular, we analyze the probability density function (*pdf*) of the extracted SOM NO3:Chl temporal patterns to identify a threshold for defining HNLC conditions ($P_{HNLC}$). We use the changes in the trend of the standard deviation calculated for each bin of the *pdf* function set the threshold ratio. To calculate

the total extent of each region (km$^2$) the spatial area of each pixel was calculated, by considering its
latitude.

*2.5 Time and space domain SOM analyses*

We use SOM (Kohonen, 1982) to elucidate spatial and temporal patterns in the complex relationship
between nutrients and phytoplankton. SOM is a subtype of artificial neural network that uses an
unsupervised machine learning algorithm to process and extract hidden structures in large datasets. The
SOM algorithm is mainly based on a training process through which an initial neural network is
transformed by iteratively presenting the input data. In this study, the architecture of the neural network
is set in a sheet hexagonal map lattice of neurons, or units, to have equidistant neurons, and to avoid
anisotropy artifacts. Each neuron is represented by a weight vector with a number of components equal
to the dimension of the input data vector, i.e. number of rows or columns in the Chl and NO3 matrices,
depending on whether the analysis is performed in the temporal or in the spatial domain. We use an initial
network composed of units of random values (random initialization). In each successive iteration during
the training process, the neuron with the greatest similarity (excited neuron), called Best Matching Unit
(BMU), is updated by replacing their values with the Chl and NO3 values of the input sample data. The
similarity is estimated by computing the Euclidean distance between the components of the input sample
and the components of the weight vector of the unit. The unit most similar to the input sample is the one
with the minimum distance. In the learning process, Chl and NO3 values of the topological neighboring
neurons of the excited neuron (BMU) are also updated replacing their values with values determined by
a Gaussian neighborhood function. In these computations, we use the imputation batch training algorithm
(Vatanen et al., 2015) where the SOM assumes that a single sample of data (input vector) contributes to
the creation of more than one pattern, as the whole neighborhood around the best-matching pattern is also
updated in each step of training. This yields a more detailed assimilation of particular features appearing
on neighboring patterns. A final neural network with the NO3:Chl patterns is obtained after repeating the
training process until a stable convergence of the map is obtained.
For typical satellite datasets, the SOM can be applied to both space and time domains. By applying the
SOM in the spatial domain, one can extract characteristic spatial patterns of the input data. If transposing
the input data matrix and applying the SOM in the time domain, one can extract characteristic temporal
patterns, i.e., the characteristic time series. Since each of these time series represents the temporal
variability of a particular region, this method can be used to identify regions of differentiated variability
on a map. The SOM, when applied to both space and time domains of the same data (called "dual SOM"
analysis by Liu et al. 2016), provides a powerful tool for diagnosing ocean processes from such different
00  perspectives. In this study we focus on the second type. We have addressed the analysis separately in the
01  time and space domains of the log-transformed NO3 and Chl datasets. In the time domain, we implement

a [4x3] joint-SOM analysis of NO3 and Chl using as input weight vectors concatenating the time-series of NO3 and Chl at each pixel, so each neuron corresponds to a characteristic joint NO3 and Chl temporal pattern over the total period of data. Since each pixel has an associated characteristic time series, we can obtain the location of a particular temporal pattern by computing the BMU for each pixel, providing a map of regions of differentiated NO3:Chl temporal variability. For the analysis herein presented only the regions with NO3:Chl>$P_{HNLC}$ are considered (regions R1 to R5).

An obstacle to the temporal domain analysis on a global scale is the opposed seasonality in both earth's hemispheres. The algorithm classifies the time series at each grid point attending to the period of the signal but does not consider time lags between the time series. Hence, pixels located either in the northern or in the southern hemisphere displaying a similar significant period in the NO3 and Chl temporal variability are classified in the same regional pattern even if they are in antiphase when the signals are seasonally lagged (6 months delayed). Regionalization is spatially coherent but the seasonal variation in the characteristic pattern that represents the neuron mixes the phenological patterns of both hemispheres. Therefore, to properly analyze the properties and trends of each of the classified regions, we have calculated the mean features of the regions by segregating the grid points corresponding to each pattern obtained from the SOM analysis into the northern, equator, and southern hemispheres (see scheme in Fig.1). Linear trends of NO3 and Chl concentrations in each region are estimated by decomposing the NO3:Chl time series in a seasonal signal plus a residual component and applying Theil-Sen slope adjustment (Sen, 1968) of the residuals of the deseasonalized series. Correlation analyses were performed using the Pearson Product Moment correlation computing best-fit linear trends using regression analysis.

The SOM analysis in the spatial domain [3x3] array, is addressed by using as input data weighted vectors consisting of spatial distributions over the global ocean of NO3:Chl ratios at a particular time. The selection of the number of neurons depends on the complexity of the data, on the features to be examined in the dataset, and on the minimization of the errors. In this case, the resulting neurons after the training loop unveil the characteristic patterns describing the spatial variability of the HNLC regions on a global scale. Then, when computing the BMU for each time we designate the extracted characteristic spatial pattern that better describes the spatial distribution of NO3:Chl ratios (P1 to P9) at each time, obtaining the time evolution of the characteristic spatial patterns throughout the considered period.

Because the SOM is based on the similarity computed from the Euclidean distance between samples, the input vectors of the different variables are normalized to the same range, before initializing the SOM computations. This guarantees a consistent comparison of the weights of the components when computing the distance of two vectors.

The size of the neural network (number of neurons) depends on the number of samples and on the complexity of the patterns and an optimal choice is important to maximize the quality of the SOM. In the present study, the map size is set to be [4 x 3] with 12 neurons for the time domain analysis, and a [3 x 3] neural network is used in the spatial domain. Using larger map sizes, the patterns are slightly more detailed, and more regions of a particular variability emerge, but the occurrence of the probability of the patterns decreases, without affecting the results noticeably (Basterretxea et al., 2018; Hernandez-Carrasco and Orfila, 2018). If a reduced neural map, such as [2 x 2] is used, patterns are concentrated together with the occurrence probability in a few rough patterns but increasing, in this case, the topological error.

SOM computations have been performed using the MATLAB© toolbox of SOM v.2.0 (Vensanto et al., 1999) provided by the Helsinki University of Technology (http://www.cis.hut.fi/somtoolbox/). Further information on SOM analysis is provided in the supplementary materials.

### 2.6 Combined SOM - wavelet coherence analysis

Joint SOM-wavelet power spectral analysis was demonstrated by Liu et al. (2016) in the study of characteristic time series of sea level variations in different regions of Gulf of Mexico. Here in this study, we expand it further to combined SOM-wavelet coherence analysis to assess the response of HNLC regions to global forcings we use an approach based on the wavelet coherence analysis (WCA) between two time-series (Grinsted et al., 2004; see Supplementary Material for further details). WCA characterizes cross-correlations by identifying the main frequencies, phase differences, and time intervals over which the relationship between the variability of HNLC regions and the main global forcings considered in this study, ENSO and MOC indexes, is strong. To do so, we first analyze the variability in both frequency and time of the characteristic time series of NO3:Chl in the different HNLC regions extracted by the time domain SOM computations and the time series of the global forcings using the continuous wavelet transform (CWT).

Cross-wavelet transform (XWT) characterizes the association between the CWT of two signals, providing information on the common power and relative phase in the frequency-time domain of two time-series. By applying the XWT to the NO3:Chl ratios and climate forcings, we determine the cyclic changes in each of the HNLC regions and their relationship with the global forcings mentioned above. Finally, we quantify the correlation between the continuous wavelet transform of two signals using the wavelet coherence analysis (WCA), In the time-frequency space the wavelet coherence coefficient $R^2$ is calculated as the squared absolute value of the smoothed cross-wavelet spectrum normalized by the product of the smoothed wavelet individual spectra for each scale (Torrence and Compo, 1998; Torrence and Webster, 1999; Grinsted et al. 2004). $R^2$ is interpreted as a localized correlation coefficient in the frequency-time

domain and it takes values between 0 (no correlation) and 1 (perfect correlation). The statistical significance level of the wavelet coherence is estimated using Monte Carlo methods as described in Grinsted et al. (2004). We use the MATLAB software package (Grinsted et al., 2004) for wavelet coherence analysis. It should be noted that cross-wavelet analysis does not establish causative relationships but only allows identifying possible linkages between variables through the synchrony of their time series.

## 3 Results

### 3.1 Global characterization of HNLC regions

The mean pattern of global ocean color data for the 20 years analyzed reveals the well-known contrast in phytoplankton biomass between the highly productive areas located in high latitudes and coastal upwelling regions, and the low-latitude oceanic waters where mean values are <0.1 mg m$^{-3}$ (Fig. 2). Low Chl regions generally correspond with low surface NO3 concentrations whereas the opposite relationship (high nitrate and high chlorophyll) is not more exceptional. Indeed, nutrient-rich productive waters are mainly restricted to shelf regions (coastal upwelling regions and shelf seas), or to the vicinity of islands (i.e. Falkland Islands) and other topographical features where multiple and overlapping sources of other elements, such as trace metals, are abundant (e.g. Boyd and Ellwood, 2010). As shown in figure 2, only in the North Atlantic, the Bering Sea, and the eastern region of the Antarctic Peninsula, Chl is enhanced. Conversely, a large part of surface ocean waters, particularly in the Southern Ocean and in the Equatorial Pacific, correspond to regions of relatively low Chl concentrations but with excess nitrate (i.e. >4 mmol m$^{-3}$).

The analysis of the normalized *pdf* of the NO3:Chl extracted from the temporal SOM analysis (shown in Fig. S2) provides a good discrimination criterion to define HNLC regions. As shown in figure 2b, the normalized *pdf* of the NO3:Chl ratio displays a marked bimodal distribution with the main mode centered at low NO3:Chl (~5 mmol mg$^{-1}$). The second mode, which corresponds to high nutrient-low chlorophyll regions, is characterized by mean and standard deviation values of $\mu$=24.1 and $\sigma$=6.7 mmol mg$^{-1}$, respectively. A critical NO3:Chl ratio bounds the lower limit of this distribution and can be estimated as $\mu - \sigma$=17.4 mmol mg$^{-1}$. Consistently, the pdf bulk analysis of its associated standard deviation (*std*) function also reveals a clear critical value located where the value of the slope varies (Fig. 2c). Both analyses allow establishing a solid statistical criterion to infer a minimum value of NO3:Chl=17 mmol mg$^{-1}$ for delimiting HNLC regions from other ocean regions. It is worth mentioning that while the *pdf* of the NO3:Chl values obtained from the SOM analysis shows a bimodal distribution, the bulk pdf of the

raw NO3:Chl values (i.e. without performing a SOM analysis) is unimodal. This suggests that the SOM technique is able to unravel relevant structures in the data that cannot be identified using classical approaches.

From the 12 characteristics time patterns of NO3:Chl variability obtained in the [4 x 3] SOM analysis, five display NO3:Chl exceeding $P_{HNLC}$ all the times (not partially) throughout the entire study period (Fig. 3a). These associated subregions (R1 to R5) match with the three traditionally reported HNLC regions (Fig.3b). In these regions surface chlorophyll rarely exceeds 0.8 mg m$^{-3}$ and the mean values range between 0.21 mg m$^{-3}$ and 0.5 mg m$^{-3}$ (Table 1). The global extent of these 5 SOM-identified HNLC subregions encompasses 25% of the ocean, being the SO by far the broadest region (18% of the ocean), whereas SNP and EEP respectively occupy some 4% and 3% of the ocean. Besides the obvious absence of HNLC regions in the northern and central Atlantic, some latitudinal asymmetries are observed in the distribution of these regions. For example, the SO region extends to lower latitudes than the SNP (i.e. ~40º S), loosely coinciding with the South Antarctic Zone limit (SAZ; Orsi, et al., 1995). Likewise, consistent with previous studies of this region (Radenac et al., 2012), the EEP displays a larger extent in the Southern Hemisphere (Fig. 3b).

The global pattern obtained from the coupled SOM analysis reflects a clear latitudinal zonation which is mainly due to latitudinal variations in nutrient availability since while chlorophyll concentration duplicates along the latitudinal gradient (R1 to R5), NO3 increases up to 7-fold (see Table 1). It is noteworthy that nutrient concentrations are generally lower in the SNP (i.e. <17 mmol m$^{-3}$) than in the SO while biomass is comparatively higher (see Table 1). Indeed, R1 in SNP only achieves the NO3:Chl criterion for HNLC regions during some periods. This region exhibits distinctive eastern and western provinces, which are consistent with previous studies describing the western region as more productive and variable (Imai et al., 2002).

Major differences among the characteristic NO3:Chl patterns in the defined subregions are not only indebted to variations in mean values but, also, to the intensified seasonal variability in higher latitudes. For example, seasonality in Chl is particularly evident in R5 and, less so, in other polar subregions (Fig. 3b). Conversely, the seasonal component of variability in the EEP is masked by the intense short-term variability.

An interesting feature depicted from the temporal SOM analysis is the positive trend in Chl experienced in the HNLC regions located in polar areas, suggesting an increase in their productivity. Decadal tendencies are in the range of 0.04 to 0.06 mg m$^{-3}$ decade$^{-1}$ in the most productive subregions (R2 to R5 in SO and R3 in SNP) but become negligible at the equator (Table 1). A regional average indicates a Chl

increase of 0.6% yr$^{-1}$ in the SNP and a 1.9% yr$^{-1}$ in the SO. In the case of the SO, positive trends are
highly influenced by a positive Chl shift occurring at the end of 2010 (see Fig. 3).

*3.2 Spatial variability of HNLC regions*

The set of 9 coherent spatial patterns resulting from the SOM analysis in the space domain and their
respective probabilities of occurrence are shown in figure 4. The organization of the maps in the figure
reveals a hierarchical classification of the maps or scenarios. Most differentiated patterns, also displaying
the highest probability of occurrence (the probability to find a pattern similar to the input data), are located
in the corners of the neural network and transitional stages connecting these scenarios fill the center. For
example, along the top and left side scenarios (P1, P2, P3, and P4), generally occurring during winter (see
Fig. 4 and 5b), the SNP extends over a larger region compared to P7, P8, and P9, which display a 3%
decrease from the mean extent. Conversely, Fe limitation in the SO, as inferred from high NO3:Chl ratios,
is markedly enhanced towards the top and right side of the figure (P4, P5, P7, P8, and P9). The extent of
the EEP region displays little variation. It should be noted that HNLC spatial extent and NO3
concentrations are not necessarily coupled since the boundaries also depend on Chl concentration values.
In addition, patterns in the proximity of the Antarctic continent are, in some cases, not well-defined during
winter due to ice cover in this region.
Figure 5 displays the time-series of the BMUs and the monthly frequency of occurrence for each pattern.
The main feature observed is the marked seasonality in the patterns shown in figure 4. The patterns with
the highest probability of occurrence, P3 and P9 (100% in April and 70% in July respectively), represent
winter and summer situations in the northern hemisphere. P4 and P8 characterize transitions toward these
patterns. Other patterns such as P6 and P2 (mostly occurring in winter and summer) are rarer but become
more frequent after 2010 (Fig. 5a). As discussed below, this variation in HNLC regional patterns (i.e. P3
substitutes P1) suggests an abrupt and major transition towards more productive HNLC regions (higher
Chl is observed).
From the nine spatial patterns shown in figure 4, we estimated the seasonal and interannual variation in
the extent of the HNLC regions (Fig. 6). Note that this regional partitioning is made on a global scale
with global criteria and therefore leads to a large-scale smoothing, which could impact the values of the
variation of the areas. However, as this signal smoothing is common to all the areas, this should not have
any effect on the regional comparison of the area variation. The magnitude of these variations remarkably
contrasts between the equatorial and polar regions. While the extent of the EEP varies by 8.9% seasonally,
changes in SNP extent can exceed 100% (Figure 6). The peak in extent for the SNP corresponds to the
boreal spring (63% of the mean value in March). In the case of the SO, seasonality is mainly driven by
changes related to the ice limit in high latitudes. Indeed, the extent of the HNLC region in the boreal
winter is 25% lower than the mean annual extent.
A remarkably good inverse correlation ($r$=-0.97, $n$=20) is observed between the interannual variations in
the extent of EEP and the SO, and a weaker though significant relationship exists between the SNP and
the EEP ($r$= -0.50, $n$=20). Therefore, as the extent of HNLC in polar regions contracts (biomass increases),
the equatorial region expands and vice versa. All three regions exhibit a shift in their extent after 2010
(Fig. 6). Both the SNP and the SO decrease after this year (5% and 2.6%) whereas the extent of the EEP
slightly increases (0.4%).

*3.3 Climate drivers of HNLC region temporal variations*
The WCA between NO3:Chl ratio in each HNLC region and ENSO are shown in figures 7a1 to a3.
Generally, small coherence structures are observed at semiannual periods; however, the main coherence
pattern corresponds to a band extending in the 2 to 4 years in the SNP and > 2yr in the EEP. This coherence
between NO3:Chl and ENSO in the 2-4 year period is particularly clear after the year 2005 when ENSO
variability intensified. In the EEP, the coherence between both series expands to periods >4 years but,
unlike in the SNP region where the NO3:Chl ratio is in-phase with ENSO signal, the signals are strongly
anticorrelated in this case (anti-phase: relative phase of 180º between both signals).
As in the case of ENSO, the MOC presents strong seasonal and interannual variations but it is also
expected to play a more active role at longer timescales (i.e. decadal and multidecadal; Buckley and
Marshall, 2016). Figure 7b shows the MOC transport index (hereafter MOI) measured at 26.5°N (Smeed
et al., 2019). Transport exceeds 17 Sv until 2009, but it weakens during 2010, stabilizing thereafter.
Generally, the MOI displays intense interannual variability, and coherence with NO3:Chl ratios is
strongest at interannual time scales (1-1.5 yr; Fig. 7b1 to b3). At this timescale, it influences NO3:Chl
ratios in the three HNLC regions yet it is more intense in the SO.

## 4 Discussion

*4.1 Global characterization of HNLC regions*
In the present study, we have addressed the analysis of the extent of the HNLC regions, their long-term
variability, and the potential drivers of these variations. Despite the relevance of precise characterization
of the extent of this biome for the estimation of the amount of carbon drawn into the ocean by
phytoplankton, objectively determining the boundaries of HNLC regions has remained elusive as it

requires coherent information from both nutrient and Chl fields. We demonstrate that a statistical approach, based on a threshold in the distribution of the global NO3:Chl ratios ($P_{HNLC}$), can robustly discriminate these regions. As in precedent studies (e.g. Moore et al. 2013), we assume that excess NO3 in surface oceanic areas is indicative of Fe limitation. This avoids relying on the scarce information available on Fe-stress or in more complex ecosystem modeling approaches. Inference of phytoplankton Fe-stress from satellite ocean color data has been attempted but it is a methodology that still presents large uncertainties (Browning et al., 2014). Furthermore, while bioavailable Fe is known to be the primary limiting factor in this relatively unproductive biome, the establishment of HNLC conditions is influenced by various other factors such as light availability, grazing pressure, rate of Fe-remineralization, and community structure, highlighting the complex interrelations among these factors. Despite these drawbacks, the herein-developed method provides results consistent with previous descriptions of the large-scale spatial patterns of HNLC regions, mostly based on NO3 fields (i.e. Archer and Johnson, 2000; Ono et al., 2008; Fu and Wang, 2022). Also, the proposed method for biome definition may introduce a bias in that the resulting spatial fields are smoother compared to those based on Fe-limitation, which is due to the greater variability of Fe concentrations compared to NO3 fields.

The $P_{HNLC}$ obtained from the *pdf* distribution of the NO3:Chl ratios represents a statistical threshold that integrates complex biological processes, including competition for resources, grazing, changes in species composition, nutrient uptake rates, Fe-regulated algal photochemistry, etc. Unlike Redfield or C:Chl ratios which respond to physiological factors within phytoplankton cells, PHNLC can be considered an environmental indicator of changes in the structure and functioning of marine phytoplankton.

According to our analysis, some 25% of the ocean (18% of the Earth's surface) corresponds to unproductive HNLC waters. With 83% of the global HNLC biome extent (Table 2), the SO is the largest region presenting clear latitudinal variation in the characteristic Chl patterns, as obtained from SOM analysis (Fig. 3). This is consistent with available descriptions of the physical and chemical properties of the SO which tend to be across latitude due to the meridional structure of the MOC and because of the rapid zonal redistribution imposed by circumpolar currents (Orsi et al., 1995). Both the SNP and the EEP respectively constitute 8% of the total HNLC extent. However, while the EEP remains relatively stable (cv=5; Table 2) the SNP can change up to 2-fold (Fig, 4 and Table 2).

Our analysis reveals marked decadal tendencies in Chl in the most productive subregions, ranging between 0.04 to 0.06 mgChl m$^{-3}$ decade$^{-1,}$ and negligible trends in the EEP (Table 1). Positive phytoplankton trends in high latitudes and no significant tendencies in the equator at the 95% interval have been previously reported by Hammond et al. (2017). Indeed, climate change projections for the 21st century predict declines in global marine net primary production but increasing Southern Ocean productivity (Hauck et al. 2015; Moore et al., 2018). Nevertheless, it is noteworthy that our analysis shows

that trends in some regions of the southern hemisphere are influenced by a Chl shift occurring in 2019-2010 (particularly R4 and R5). This non-linear enhancement in phytoplankton, which is not exclusive to oceanic Fe-limited waters (see for example Marrari et al., 2017), positively biases the Chl increase rate in these subregions. It is unlikely that this change in overall phytoplankton biomass is related to satellite data merging. SeaWiFS ended operations in 2010 and MERIS sensor ceased in April 2012; however, a decrease in biomass would be expected from these changes (Garnesson et al., 2019). The possible source of this shift in phytoplankton biomass is more extensively discussed below.

### 4.2 Spatial variability of HNLC regions

SOM analysis allowed the characterization of the seasonal and interannual variability of HNLC regions (Fig 5 and 6). The SO and the SNP are regions of seasonal extremes in productivity as a consequence of the large fluctuations in the environment that they experience. Important seasonal variability is observed in the SO, which is attributed to light limitation during winter and to variations in iron stocks occurring in surface waters (Tagliabue et al., 2014). Therefore, while presenting HNLC characteristics for most of the year, the SO exhibits distinct Chl variability patterns that are well captured in the SOM regionalization and the characteristic patterns of each subregion. In the case of the SO, the changes in the extent are reduced (20% from mean values) which is attributed to the large-scale nature of the physical processes regulating the productivity of this region (Cullen, 1991). Mid-depth and deep ocean waters communicate with the ocean surface after following long a circuit route driven by ocean overturning circulation (Lumpkin and Speer, 2007). As reported by Smeed et al. (2018), the variability of the MOC flow system has an important decadal component associated with thermohaline forcing. This long-term component of variability could dominate over higher frequency variability but the length of the observational record of the AMOC is still insufficient to resolve variations at this scale.

Variability patterns in the SNP, are attributed both to marked seasonality in the local forcings and to fluctuations in the regional circulation patterns and the consequent advective Fe enrichment processes from river discharges of the surrounding marginal regions in the North Pacific (Cummins and Freeland 2007; Takeda, 2011; Lauderdale et al., 2020; Nishioka et al., 2020). In the Northern Pacific, surface Fe availability is driven by vertical mixing processes which closely relate to seasonal variations in weather conditions (Nishioka et al., 2007; 2020). Upwelling is triggered by severe winter cyclones that generate enough Ekman pumping to maintain high nutrient concentrations in the near-surface waters (Gargett, 1991; Harrison et al., 2004).

Marked differences in variability are observed between the subpolar regions and the EEP, where seasonality is marginal. Nevertheless, it should be noted that the EEP is a peculiar region that integrates

subregions with 6-month out-phased seasonal variations (north and south of the equator). On average, the south-equatorial region contributes more (62% of the total extent) to the mean extent of the EEP whereas the Northern subregion determines most of the observed variability (Fig. S3).

At interannual scales, the extent of HNLC regions is more modest (up to 5%; Figure 6) but all three regions are notably correlated, suggesting that their interannual variations are driven by global-scale processes. In particular, the SO and the EEP are highly correlated ($r=0.97$). This coupling between Southern Ocean productivity, and equatorial productivity, was suggested by Dugdale et al. (2002) based on the observed nutrient ratios. They proposed that both regions were connected, out of phase, by the meridian SubArctic Mode Water. Nevertheless, the seaways in the Pacific are complex, and determining the overturning signature is challenging. Some studies have suggested that temperature anomalies subducted into the pycnocline at subtropical latitudes may not reach the Equator with any appreciable amplitude (Schneider et al., 1999a). However, mass water balances in the equatorial pacific reveal that the strength of the equatorial upwelling is related to variations in the Pacific overturning (PMOC) (McPhaden and Zhang, 2002), and therefore, an influence on EEP extent is expected. Likewise, meridional circulation extends to the North Pacific; however, the SNP does not ventilate the deep ocean at significant rates and the PMOC cell at this latitude corresponds to a rather independently functioning intermediate water cell (Warren et al., 1983) . MOC in this region is reportedly weak (1-4 Sv) and extends no further than 50ºN (Ishizaki, 1994; Yaremchuk, 2001). However, there is evidence showing the response of this region to changes in PMOC (Burls et al, 2017, Holzer et al, 2021). For example, it has been observed in TOPEX altimeter data that MOC influences the basin-scale baroclinic circulation in the SNP (Kuragano and Kamachi, 2004). This would explain the lower, yet significant, correlation with the variations at the SO and EEP. Indeed, the shift observed in 2009-2010 is common to all three regions (albeit with a different sign).

The causes of the drastic 2009-2010 variation in the extent of HNLC regions are uncertain but they can plausibly be related to changes in the strength of meridional circulation and the consequent atmospheric and oceanic variations. For example, Moat et al. (2020) suggest a previously unsuspected role for the AMOC in climate variability during the 2010 event which coincided with a cold winter in Europe. Several ocean scale changes that may be related to the observed changes in the HNLC region extent have been reported in the period 2009-2011. For example, rapid warming, salinification, and a concurrent dissolved oxygen decline have been observed at BATS during the 2010s (Bates and Johnson, 2020). There is also evidence indicating that a decadal intensification of Pacific trade winds weakened in 2011 (Bordbar et al., 2019). Trade wind intensity variations in the equatorial pacific region are associated with SST anomalies, weakening of the equatorial divergence, and changes in upper-ocean thermal structure (England et al., 2014; Bordbar, et al., 2017). The relationship between the equatorial wind intensity and

the equatorial undercurrent strength is also well established (McPhaden, 1993). These atmospheric changes, affecting the upwelling of Fe in the EEP (and indirectly to other oceanic regions), agree with the hypothesis of Winckler et al. (2016) suggesting that ocean dynamics, not dust deposition, control the equatorial Pacific productivity. In our case, we observe a reduction of the extent of the HNLC region in the EEP during an enhanced wind intensity period (before 2011) and an expansion thereafter (Fig. 6). Connections of this mechanism to high latitude HNLC regions reveal large-scale adjustments with consequences in global ocean productivity.

### 4.3 Drivers of HNLC region variability

ENSO is the primary source of the interannual variability in this region and its occurrence is related to the decline in NO3 supply. Sub-decadal fluctuations in Chl in the EEP region displaying a good correlation with the ENSO index have been reported before (Oliver and Irwin, 2008; Boyce et al., 2010). By contrast, the SO only shows weak evidence of this relationship which suggests that ENSO is not a major forcing driving the variability of NO3:Chl in this region. This is consistent with reports from Ayers and Strutton (2013) who did not find a significant relationship between nutrients in this region and ENSO events. Similarly, Racault et al. (2017) reported evidence indicating that during Eastern Pacific and Central Pacific types of El Niño events, impacts on phytoplankton were widespread, but tended to be greatest in the tropics and subtropics, encompassing up to 67% of the total ocean affected areas.

It can be argued that differences in the response to ENSO are due to the different nature of the forcings driving nutrient supply in each region. While the EEP and the SNP seem to respond to ENSO-related changes in wind forcing, NO3:Chl ratios in the SO are more stable and respond to annual and semi-annual variations. The coupling between the EEP and the SNP dynamics has been reported before. Qiu (2002) observed progressive shoaling of the Alaska gyre caused by a strengthening of the cyclonic circulation. The interannual variability of this gyre was connected to ENSO-related equator-originated sea surface height anomalies. Several large-scale climate pattern indexes are invoked to explain physical and biological fluctuations in the SNP. For example, Di Lorenzo et al., (2008) defined the North Pacific Gyre Oscillation (NPGO), which nicely explains the fluctuations of salinity, nutrients, and chlorophyll related to the circulation in the North Pacific gyre. It is beyond the scope of the present work to assess the relationships of all these indexes with the variability in the extent of HNLC regions. Nevertheless, the proposed climate indexes for the Pacific present a high relationship among them, which highlights the strong dynamical linkages between tropical and extratropical modes of climate variability in the Pacific basin, and the important role played by ENSO (Di Lorenzo et al., 2013).

A stronger MOC should result in the upwelling of macronutrients and Fe at faster rates as well as in increased Ekman transport of nutrients equatorward and subsequent subduction (Ayers and Strutton, 2013). This is observed in the high positive cross-wavelet correlation at 1-1.5 yr in the EEP region. In addition, a clear variation in the coherence phase is observed, being 90º (3-4 months) in the SNP, in phase for the EEP, and 270º (9-11 months) out of phase in the SO, suggesting a meridional propagation of the MOC effect. Figure 8b also reveals a decline of the MOC until 2010 that has remained ~ 15% below its pre-2010 level since then (~17 Sv; Ayala-Solares et al., 2018; Caesar et al., 2018). This trend has been attributed to climate warming and the consequent changes in the hydrological cycle, including sea-ice loss and accelerated melting of the Greenland Ice Sheet, causing further freshening of the northern Atlantic (Bakker et al., 2016; Böning et al., 2016). It has been proposed that AMOC weakening will affect large regions of the world's upper oceans that are currently supplied with nutrients by the South Antarctic Mode Water (Schmittner, 2005).

Weakened MOC after 2010 and, the particularly low value during that year (Fig. 7b), is coincident with the shift in the extent of the HNLC regions shown in Figure 5, suggesting that weaker MOC is related to increases in the extent of the EEP and contraction of the SNP and the SO HNLC regions. In the case of the EEP, it could be proposed that a larger equatorial area with Fe deficiency would be associated with a decline in upwelling intensity. It is estimated that the slowing down of the overturning circulation in the Pacific Ocean since the 1970s has generated a decrease in upwelling of about 25% in an equatorial strip between 9° N and 9° S (McPhaden and Zhang, 2002). Nevertheless, the larger HNLC region in the EEP could also be explained by the depletion of Fe in the source waters feeding the EUC, as reported by Kaupp et al. (2011).

At high latitudes, the weakening of the AMOC, is coherent with a decrease in the extent of the SNP and the SO (Fig. 5). This anomaly, starting in 2009-2010, is a global feature also reflected in the intertropical convergence zone (ITCZ) time series, an atmosphere's energy balance indicator (see Green et al., 2017; Ibanhez et al., 2017), suggesting a strong atmosphere-ocean coupling with impact on ocean productivity. It is not clear how a reduced flow would favor the increase in biomass in high latitudes. While it is plausible that in the case of abrupt and/or permanent variations of AMOC primary production, and hence phytoplankton biomass, will be reduced, it is less clear how present-day variations influence phytoplankton biomass. It has been proposed that a reduced AMOC from increased precipitation and melting sea ice, could contribute to reduce vertical mixing which may increase productivity in polar regions (Riebesell et al., 2009). Other studies (e.g., Martinez-García et al., 2009), showed a relationship between AMOC and Chl variations, mainly due to the interaction of the main pycnocline and the upper ocean seasonal mixed layer. In addition, some paleoclimatic studies have demonstrated that AMOC weakening can increase the productivity north from the Polar Front, but only if an increase in the

atmospheric soluble Fe flux is considered (Muglia et al., 2018). Paleoceanographical records reveal a strong correlation between proxies of aeolian Fe flux and productivity has been reported in this region (Kumar et al., 1995; Martínez-García et al., 2009) but, in present times, dust deposition in this area has notably varied and this effect is unlikely to be important at the time scales considered here. Complex ecosystem processes including competition for Fe with bacteria, Fe remineralization rates, and organic complexation processes could determine the phytoplankton response under future scenarios. Further, biomass building up is not only driven by nutrient availability. Changes in biomass can be produced by variations in the thermocline depth affecting the vertical distribution of phytoplankton. Nevertheless, variations in phytoplankton composition, physiological adjustments in cellular pigmentation, and grazing could also modulate Chl variability. Indeed, the prevailing foodweb structure may play an important role in Fe fertilization (Schmidt et al., 2016). At larger scales, there are still unresolved questions about the couplings occurring at different temporal scales. For example, MOC variations are known to interact with ENSO variability (Dong et al., 2006; Dong and Sutton, 2007; Timmermann et al., 2007). These connections provide further evidence of the global scale coupling and feedback between the atmosphere, the ocean, and global productivity variations.

## 5. Conclusions

Variations in the boundaries of the HNLC regions can provide an integrative view of how climate scale ocean variations influence ocean productivity. We established a statistical criterion to identify HNLC regions from global Chl and NO3 data that sets the basis for systematic analyses of HNLC regions and their response to climate variations.

Our results suggest that while local-scale processes can determine certain aspects of the productivity of HNLC regions, their long-term patterns are strongly influenced by variations in global atmospheric and oceanic circulation. We observed significant interannual variations in the extent of HNLC (up to 5% in Fig. 6), which are associated with anomalies in global forcing intensity. Accordingly, our findings suggest a scenario in which HNLC regions are vulnerable to interbasin teleconnections rather than local forcings. These general patterns may be modulated by feedback between different forcing mechanisms. For instance, there is a positive correlation between PMOC variability and El Niño-Southern Oscillation (Tandon et al. 2020). Furthermore, our analysis reveals a shift in phytoplankton biomass and HNLC variation patterns occurring at the end of 2010, which evidences the occurrence of fast transitions in ocean biogeochemistry. The underlying drivers of these regime shifts and the resulting biological responses to these ocean-scale changes require further investigation since they are a fundamental aspect of long-term variations in marine ecosystem functioning.

Finally, the present study highlights the importance of maintaining long and coherent datasets beyond satellite-borne information to be able to disentangle the different components of variability, particularly at long timescales, and to evaluate the impact of climate change on marine ecosystems. Most of the geochemical information at this scale (i.e. nutrient and Fe fields) will probably require further global sampling programs and refined modeling.

## Author contributions

Data were processed and analyzed mainly by G.B., J.S.F, and I.H-C. Writing by G. B., J.S.F, and I. H-C. and S.A.S. The authors declare no competing financial interests.

## Funding sources and data references

This work was partially supported by SIFOMED grant (CTM2017-83774-P) from Ministerio de Ciencia, Innovación y Universidades, the Agencia Estatal de Investigación (AEI), and the Fondo Europeo de Desarrollo Regional (FEDER, UE). G. Basterretxea was supported by Salvador de Madariaga PRX18/00056 scholarship. J.S Font-Muñoz acknowledges funding by an individual postdoctoral fellowship "Margalida Comas" (PD/018/2020) from Govern de les Illes Balears and Fondo Social Europeo. I. Hernandez-Carrasco acknowledges financial support from the project TRITOP (grant UIB2021-PD06) funded by Universitat de les Illes Balears and FEDER (EU).

All data included in the present study is accessible from the following publicly available repositories: CARINA (https://www.nodc.noaa.gov/ocads/oceans/CARINA/), COPERNICUS (https://marine.copernicus.eu/), ECMWF (https://www.ecmwf.int/), GEOTRACES (https://www.geotraces.org/), GlobColour (www.globcolour.info), MERCATOR (https://cmems-resources.cls.fr/), NERC (https://nerc.ukri.org) and NOAA ( https://www.nodc.noaa.gov/).

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

## Tables



Table 1. Mean±std characteristics of each of the SOM-defined HNLC subregions (R1 to R5). NO3 (mmol
m$^{-3}$ ) and Chl (mg m$^{-3}$) values are respectively from model and satellite data. Decadal chlorophyll trends
($\Delta$Chl, mgChl m$^{-3}$ decade$^{-1}$) are calculated from the mean time-series of monthly deseasonalized
chlorophyll.

| Region | Subregion | NO3 ($\mu$M) | Chl (mg m$^{-3}$) | NO3:Chl (mmol NO3 mg Chl$^{-1}$) | $\Delta$Chl (mg m$^3$ decade$^{-1}$) |
|---|---|---|---|---|---|
| SNP | | | | | |
| | R1 | 4.51 ± 1.02 | 0.31 ± 0.07 | 15± 3 | +0.05 |
| | R2 | 8.05 ± 0.88 | 0.36 ± 0.07 | 23± 6 | +0.26 |
| | R3 | 15.52 ± 2.27 | 0.49 ± 0.16 | 35 ± 15 | +0.43 |
| EEP | | | | | |
| | R1 | 4.04 ± 0.77 | 0.22 ± 0.02 | 18 ± 3 | +0.01 |
| | R2 | 6.63 ± 1.42 | 0.39 ± 0.05 | 20 ± 4 | +0.08 |
| SO | | | | | |
| | R1 | 4.13 ± 1.05 | 0.22 ± 0.06 | 20 ± 4 | +0.24 |
| | R2 | 9.11 ± 1.23 | 0.31 ± 0.06 | 31 ± 9 | +0.42 |
| | R3 | 15.73 ± 1.07 | 0.32 ± 0.10 | 55 ± 17 | +0.47 |
| | R4 | 23.26 ±1.06 | 0.26 ± 0.16 | 104 ± 32 | +0.62 |
| | R5 | 29.18 ± 1.57 | 0.43 ± 0.92 | 103 ± 54 | +0.46 |



Table 2. Basic statistics of the extent of each of the SOM-defined HNLC subregions during the
analyzed period (1998-2017).

| Region | Mean extent±S.D. (x10$^6$ km$^2$) | % of total extent | Min-Max (x10$^6$ km$^2$) | Max. Variation (x10$^6$ km$^2$) | C.V.% |
|---|---|---|---|---|---|
| SNP | 7.7±3.6 | 8.4 | 3.8-15.9 | 12.1 | 47 |
| EEP | 7.8±0.4 | 8.4 | 7.3-8.4 | 1.1 | 5 |
| SO | 76.5±0.9 | 83.2 | 61.3-86.8 | 25.5 | 12 |



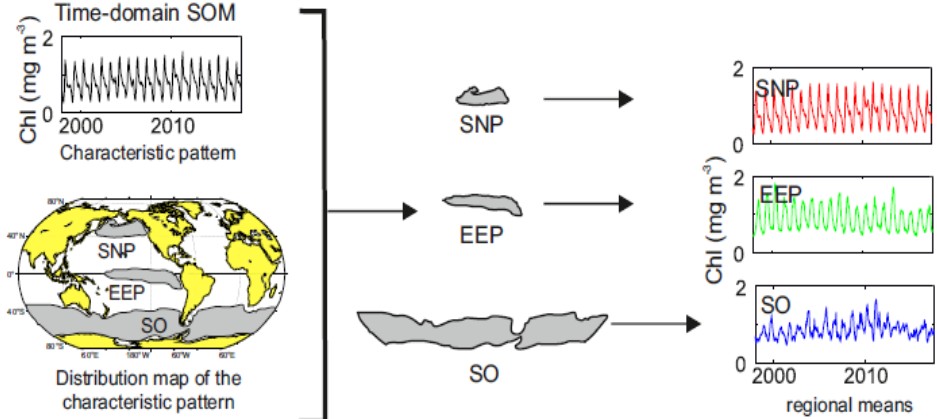


Figure 1. Scheme of a characteristic pattern and its distribution map obtained from SOM time-domain
analysis at global and regional means calculated for each of the HNLC regions.



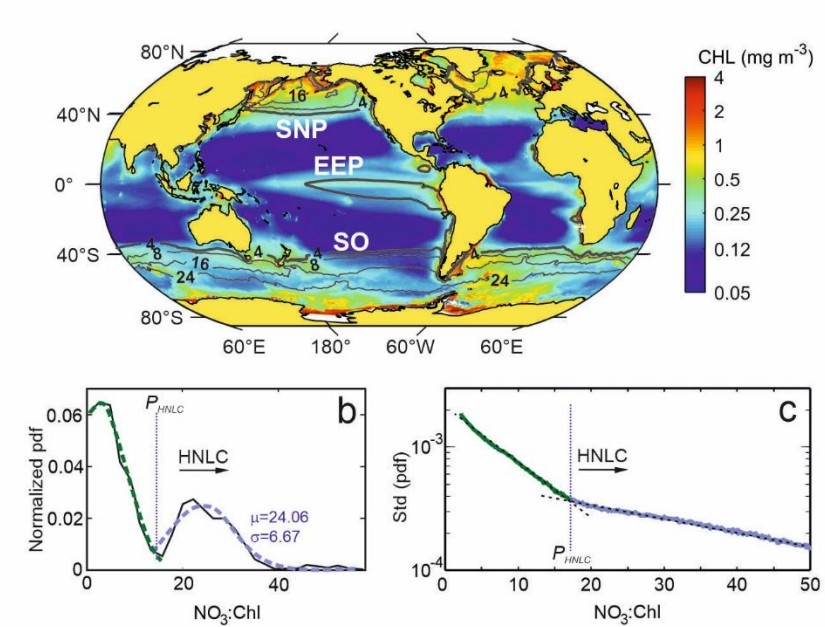

Figure 2. a) Global mean satellite Chl concentrations (mg m$^{-3}$) for the period 1998-2018 and superimposed surface NO3 contour lines (from modeling data (isolines are drawn at 4 mmol m$^{-3}$ intervals). b) Normalized probability density function of the values of the NO3:Chl ratio obtained from the SOM temporal patterns. Green and blue lines show the fit to a normal distribution for the first and second pdf modes, respectively. c) Standard deviation of the probability density function (*pdf*) of the NO3:Chl (mmol/mg) monthly ratios obtained for the 20 years analyzed. Note that the y-axis scale is logarithmic. The critical point ratio $P_{HNLC}$ =17 mmolNO3 mgChl$^{-1}$ delimits HNLC regions from macronutrient limited regions.

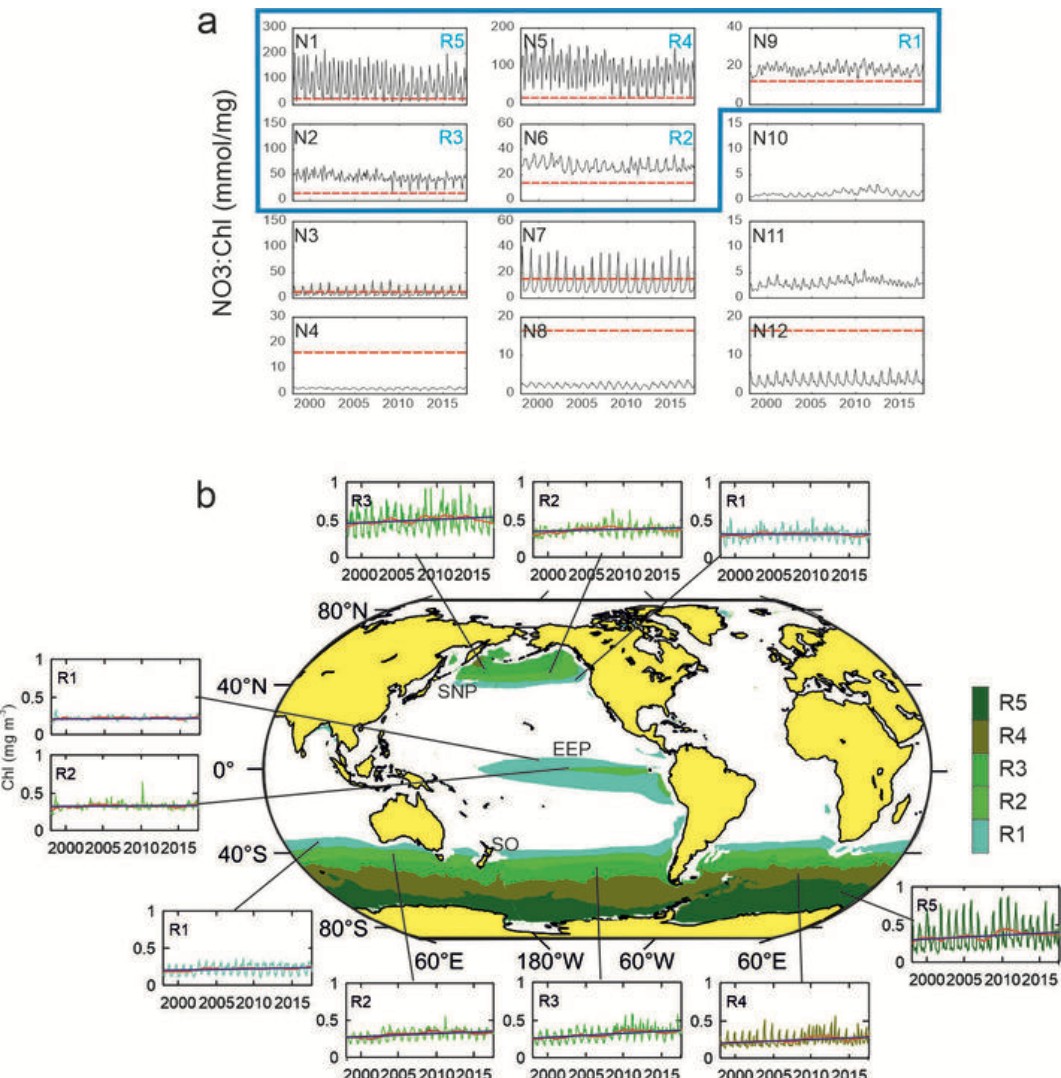

Figure 3. a) Characteristic temporal variability patterns of NO3:Chl ratios (N1-N12) as unveiled by the 4x3 SOM analysis in the time domain. Red dashed lines indicate the $P_{HNLC}$ value  b) Coherent regions of HNLC variability (SNP, EEP, and SO) and corresponding subregions (R1 to R5)  associated with the SOM temporal patterns exhibiting only NO3:Chl values larger than $P_{HNLC}$ all the times throughout the entire analyzed period (i.e. identified with N9, N6, N2, N5 and N1) . Patterns corresponding to a subregion in the northern and southern hemispheres present a similar pattern although seasonally lagged (6-month delay). Insets show the time series of the averaged Chl over the corresponding subregion (complete map of regions of NO3:Chl variability and corresponding temporal variability patterns  are shown in Fig. S1). The red line represents the 24-month filtered series and the blue line indicates the trend (values shown in Table 1).

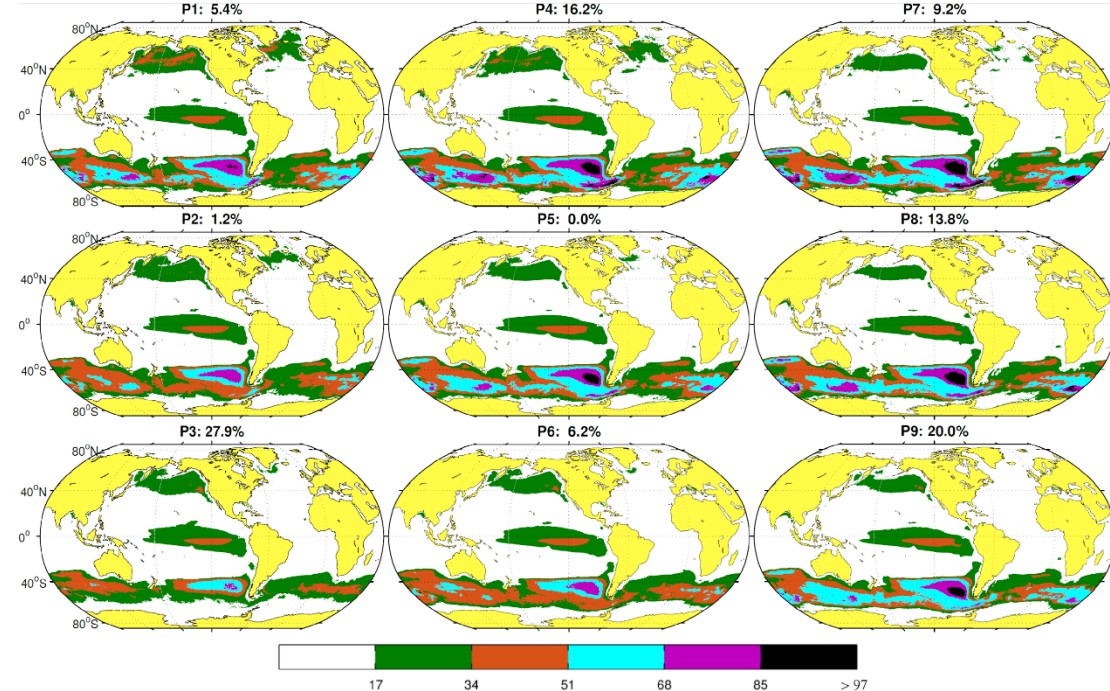

Figure 4. Characteristic spatial patterns (P1 to P9) of HNLC regions as defined by NO3:Chl ratios > $P_{HNLC}$. The value on top of each pattern indicates its probability of occurrence over the 20-year period analyzed. To preserve the topology, the SOM algorithm introduces some patterns with zero probability of occurrence, such as P5. The colorbar indicates the different NO3:Chl ranges represented.

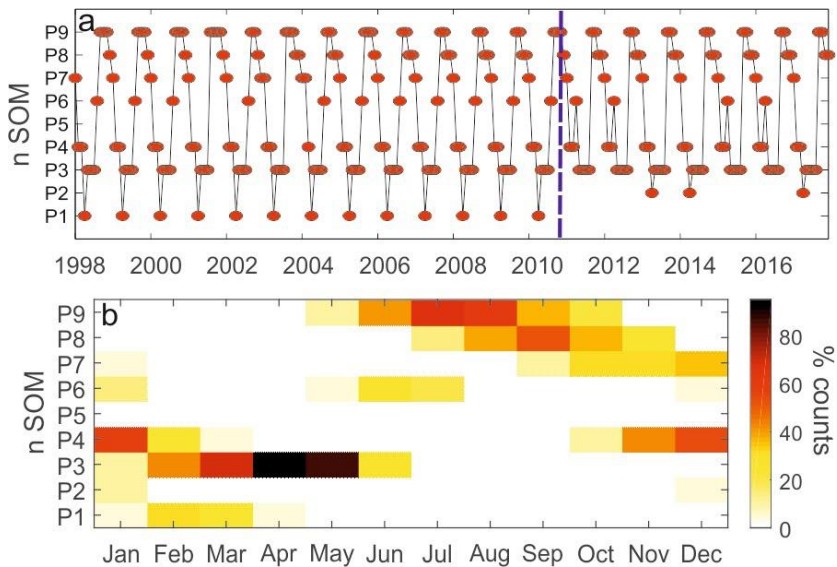

Figure 5. a) Time evolution of the spatial patterns as defined by the Best Matching Units (BMUs) for the period of 1998-2018. The blue dashed line indicates the regime shift occurring after 2010. b) Monthly frequency of occurrence of the spatial patterns identified in Figure 4.

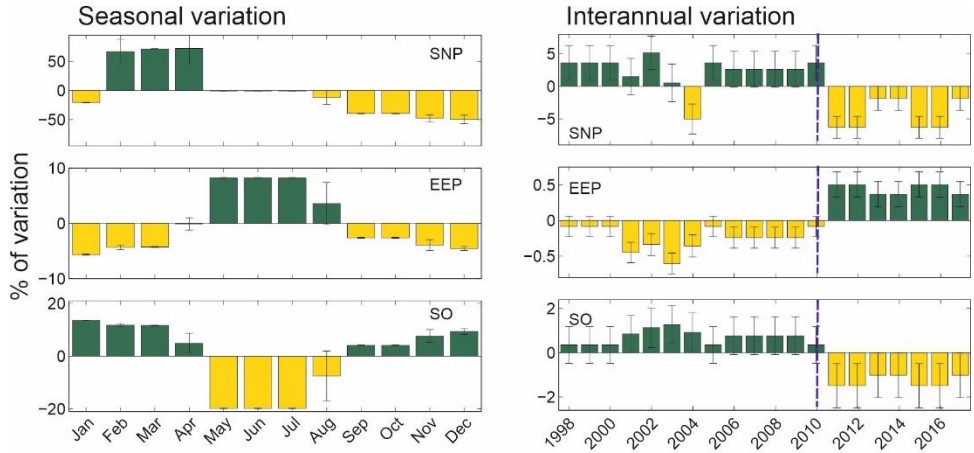

Figure 6. Seasonal (left) and interannual variations (right) in the spatial extent of the three HNLC regions, represented as a percentage of variation from the mean extent of each region. Variations are referred to the mean extension of each region. Dark and light-colored bars indicated positive and negative values, respectively. The blue dashed lines indicate the regime shift occurring after 2010.

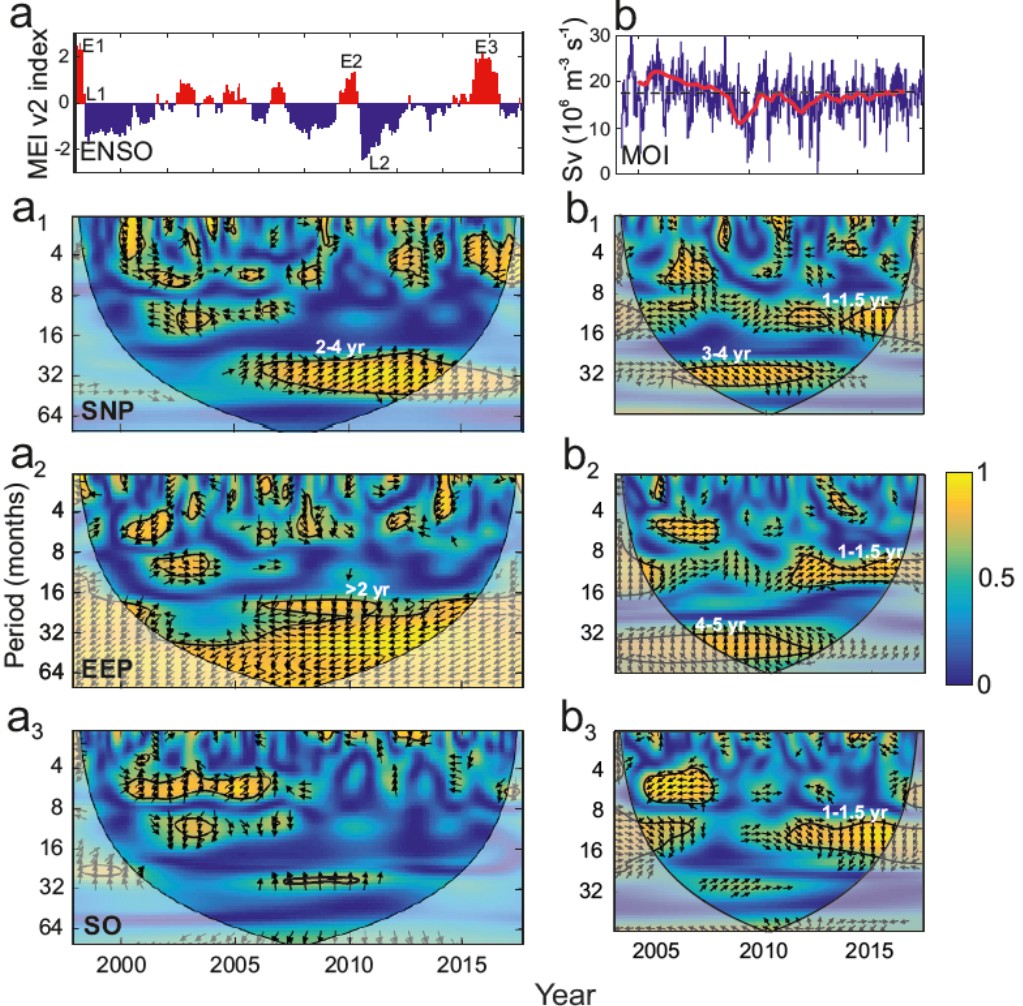

Figure 7. a) ENSO (MEI v2) index. E1-E3 indicate intense El Niño episodes and L1 and L2 mark strong La Niña periods. a1, a2, a3 display the cross-wavelet coherence between NO3:Chl ratio and ENSO for each of the HNLC regions. b) Meridional overturning (AMOC) volume transport for the period (2004-2017) measured at 26.5°N (Smeed et al., 2016). The red line shows interannual component is obtained by filtering the data with a 540-day low-pass filter after the removal of the mean seasonal cycle. b1, b2, and b3 display the cross-wavelet coherence between NO3:Chl ratio and AMOC for each of the HNLC regions. The thick black contours in the cross-wavelet coherence figures designate the 95% confidence levels and the cone of influence where edge effects are not negligible is shown as a lighter shade. The arrows indicate the phase relationship between the signals with the horizontal component indicating in-phase (rightward) or out-of-phase (leftward) and the vertical component indicating a 90º phase difference lagging (upward) or leading (downward). Period units are months.