# Peer review of "Global variability of high nutrient low chlorophyll regions using neural networks and wavelet coherence analysis."

_EGUsphere, 2022_

## Referee Comment (RC2)

This paper relies on the spatio-temporal variability of the HNLC regions identified by SOM over the three major ocean areas using satellite-derived chlorophyll-a and modeling outputs of nutrients. The authors have performed the NO3:Chl as an indicator of the distribution limit of HNLC. They have demonstrated the linkages between HNLC extent and some climate-driven factors and teleconnections.

As a first very general comment, I would say that this is a valuable case study that can be published with some major corrections. The authors presented a lot of data and analysis procedures that need accurate processing schemas and precise interpretation, which they handled well.

The introduction section is well written and presents an adequate understanding of the presented work. The methodologies are adequate, but need some improvements. Some supplementary materials may be inserted into the main text because of their essential investigations and frequent references to them (e.g., Fig. S2).

The spatio-temporal variability of HNCL/NO3:Chl should be presented more precisely and quantitatively. For example, maps of monthly climatology, charts and maps of inter-annual cycles, and spatial-temporal cross section such as Hovemoler chart may present valuable results.

Some of the findings of the results section have not been well demonstrated in this study (e.g., section 3.4.1 Influence of SST variations – the global power spectra of CWT are needed to explain the intra-annual and inter-annual cycles). Some contents of Figures need to be better framed/explained. Some words would be expressed in accurate form. e.g., it is not clear in many places in the text that the "wind" word is mean wind speed, or wind vectors, or wind stress, or wind components (zonal and meridional).

I think a discussion section is required to explain the performance and limitations of the presented methodologies and results, which are not seen in this manuscript.

Bioregionalization analysis was used in previous studies to classify the global oceans (e.g., Longhurst, A. (2007), Ecological Geography of the Sea, Academic Press, London). Could the author highlight the need to new ocean's regionalization which are not accessible from available global regionalization? And why SOM and not the other classification methods such as k-means?

I think the findings of nutrient model are not presented well. Some additional information is required.

The Mixed Layer Depth (MLD) is one of the main oceanographic indicators that can be used for interpretation of nutrients and phytoplankton variabilities. Does the nutrient model consider the MLD? If yes, please indicate in the text. And if not, I think it is required to consider the global ocean MLD in your work.

Looking to be constructive, in addition to the overall comments above, which should be taken into account in a possible review, I would like to point out the following Remarks.

1- In the 2.1 Ocean color data:

The GlobColour data are presented in 25 km spatial resolution globally. The authors have mentioned that the composites have a 0.25° spatial resolution. We know that the spatial resolution of 25 km and 0.25° are not the same specially at higher latitudes. If the 0.25° is true, please explain the spatial interpolation methods. If not, please correct.

2- In the section: 2.2 Nitrate data

The authors have made some essential assumptions that need to be approved precisely. May be explain more in the Results.

3- Please provide more information about the setup of the SOM algorithm (which are available not in the text nor supplementary material), in particular about the initial configuration: linear or random initialization, sheet or toroidal network, etc. Which neighbor function (Gaussian, Ep, et) is used to update the neighbors of the excited neuron (BMU) after each iteration during the training process.? Did the authors check the sensitivity of the SOM pattern to linear and random initialization?

4- The output of the SOM is also not well defined beyond being a map or topology; for example, how are the errors computed? The number of neurons is chosen not only depending on the topological errors (or topographic errors), but also on quantization errors. How different are the patterns when the number of neurons is, for example, 9?

5- Figure 6 and 7. The arrows are too small.

6- I recommend to show the time-series anomaly of HNLC and teleconnections at the top of the Fig. 6 and indicate the significant inter-annual cycles.

7- It is hard for readers to infer the shift after 2010 (Fig. 5). I think more visualization/explanations of data are required.

8- There are some abruptions in the significant annual cycles (1.5 year, from year 2006 to 2010) in the Fig. 7 which need to be explained. I suggest perform the global spectral cycles graphs.

9- The authors have considered SST and the teleconnections as a factor controlling HNLC variability. Are there any environmental factors such as precipitation and wind stress that may affect HNLC variability? Please discuss.

---

## Author Comment (AC2)

**RC1**: 'Comment on egusphere-2022-827', Anonymous Referee #1, 25 Oct 2022  reply

This paper applies several novel statistical methods to assess the extent and variability of the ocean's three major "high-nutrient / low-chlorophyll" (HNLC) regions. Its novel contributions include articulating a new definition of HNLC in terms of the ratio NO3/Chl, and identifying new linkages between HNLC extent and some climate variability indices.

We would like to sincerely acknowledge RC1 for the outstanding review of this manuscript. It is rare these days to receive such detailed and appropriate feedback. His/her comments have undoubtedly contributed to a significant improvement of the manuscript. We highly appreciate his/her dedication.

I think this is a good paper that is publishable with revisions. The English is mostly good although there are some quirks. However, I lean toward major rather than minor revision for this reason: some papers that combine Results and Discussion cry out for the separation of the two, and this is one of them. I think it would be much stronger if it were rewritten in the standard I-M-R-D format. Start with: What did we learn and what is the evidence? (Results) and then: What does it mean in the context of the existing literature and potential future research? (Discussion). At present, the results of this research are mixed up with speculation and literature review in a way that detracts from the paper's core messages and makes it difficult for the reader to identify what exactly the research conducted demonstrates. There are also passages in the Methods that I think more properly belong in the Discussion (e.g., 124-129).

We have addressed most of the issues suggested and thoroughly reviewed English grammar. Admittedly, we were reluctant to rewrite the R&D section in the standard I-M-R-D format because the main text could become excessively long and, eventually, repetitive. This problem is common to many papers based on satellite information in which detailed descriptions of spatial fields of different variables are made. However, following the reviewer's suggestion we have segregated results from the discussion and we recognize that, in this case, it may be a good option. We have also trimmed the conclusions section to highlight the fundamental points of the manuscript, and, as the reviewer suggests, we have moved lines 124-129 to the Discussion section.

I also do not think that the statistical methods are adequately explained. On 182 we have "Statistically significant trends were considered those exceeding the 95% confidence level." This would seem to be a straightforward significance test of simple linear regression. But even here some more detail is needed, e.g., what is the decorrelation time scale and therefore the effective sample size? (see e.g., www.sciencedirect.com/science/article/pii/B978012387782600003X) When we get to the CWT (which term appears only twice and is defined differently each time), we are simply told that "The thick black contour designates the 95% confidence level" (Figure 6 caption). The text says nothing about how the confidence level is calculated (see also Figure 7). It is stated in the Supplement that "Monte Carlo simulations based on two uniform white noise time series are used to determine the significance level", but more detail is required and this should be at least mentioned in the main text. Ultimately, the variation in the total HNLC extent is found to be on the order of 5% (446). How do we know this is significant and not just noise? The coherence across regions and with the climate indices suggests that it is, but the lack of clarity regarding methods, and the sometimes too-good-to-be true correlations (see next paragraph) detract from the presentation.

We have improved some methodological descriptions. Specifically, linear trends were determined by Theil-Sen slope adjustment [Sen, 1968] of the residuals of the deseasonalized series. The Theil–Sen estimator is a well-known estimator of the true slope based on non-parametric that has low sensitivity to outliers, by taking the median of all possible slopes between pairs of data, instead of the mean. The significance test for the slope is a test for the trend. Correlation analyses were performed using Kendall's tau correlation coefficient.

The CWT (Continuous Wavelet Transform) is a signal processing and data analysis technique generally used to decompose a signal into different time scales and analyze its frequency content at each scale. This is achieved using functions called "wavelets", which are waveforms that can expand or contract in time to adapt to different scales. The CWT must first be applied to the signal to obtain a representation of the signal in the wavelet space. The amplitude or intensity values at each time scale can then be calculated. The significance of the CWT is assessed by comparing the background power spectrum with Monte Carlo generated red noise. The Monte Carlo method is used to generate a set of simulated data that has the same statistical characteristics as the original signal. This is done using a known probability distribution function, such as a normal or uniform distribution.

Finally, the amplitude values of the original signal are compared to the amplitude values of the simulated data to determine whether they are statistically significant or not. This is done using statistical tests, such as the Student's t-test or the Wilcoxon test. If the amplitude values of the original signal are statistically significant compared to the amplitude values of the simulated data, it can be concluded that there are specific patterns or characteristics present in the original signal that are not simply the result of noise. Using shaded regions to represent the level of significance in the final figure of the CWT analysis is a visual way to indicate which amplitude values are statistically significant and which are not.

To analyze the correlation between the continuous wavelet transform of two signals we compute the cross-wavelet coherence analysis (CWA), which has been properly mentioned in the new version of the manuscript. We skipped this sentence in the previous version of the manuscript. The significance of the wavelet coherence is also assessed using Monte Carlo methods, but now generating pairs of red noise data. We have improved the description of CWT and the CWA adding part of this discussion in the M&M Section.

The estimation of the interannual variation of the HNLC areas is based on the spatial patterns of HNLC obtained from the SOM analysis applied to global ocean data. This regional partitioning is made on a global scale with global criteria and therefore leads to a large-scale smoothing, which could impact the values of the variation of the areas. However, as this signal smoothing is common to all the areas, this should not have any effect on the regional comparison of the area variation.

In 3.3 we have "The relationship observed in interannual variations in HNLC areas suggest a global scale coupling between the equator and the poles. Good inverse correlation (r=-0.99, n=20) is observed between the interannual variations in the extension of EEP and the SO, and a weaker thought significant relationship exists between the SNP and the EEP (325 r= -0.75, n=20). Therefore, as the extension of HNLC in polar regions contracts (biomass increases), the equatorial region expands and vice versa. All three regions exhibit a shift in their extension after 2010 (Fig. 5)."
I find a correlation of -0.99 difficult to credit. But when I look at Figure 5 the SO and EEP interannual time-series do indeed look like mirror images of one another. But how is this possible? What physical process could account for it? This is the kind of result that readers will dismiss without more attention to detail. But the discussion following this result is vague and speculative. I think the authors need to work harder at explaining how such a tight coupling could exist and convincing the reader that it is not just an artifact of their analysis method.

Thanks for pointing out this seemingly strange correlation which may raise doubts about the calculations. We have thoroughly reviewed the calculations and we stand that it is not an artifact or an error. However, the extent variability estimations were performed based on the number of pixels considered HNLC according to the criteria established in Section 3.1. We have recalculated the values of the areas in terms of actual surface (Km2), instead of the number of pixels, by considering the latitudinal variation in pixel extent. Figure R1 shows the results of these computations. While maintaining the general pattern, interannual variations are slightly different and, indeed, the correlation between NPP and EEP, and between SO and EPP reduces to - 0.49, -0.97. Beyond this correction, the correlation between SO and EEP extent is still outstandingly good, which suggests that both systems respond to a common forcing. Both ENSO and, as explained below, MOC are common to the EEP and the SO, and less so to the SNP. We discuss this in the new version of the paper. However, a clear mechanistic understanding of the coupling between the EEP and the SO would require information beyond the scope of the present study.

[Figure]

[Figure]

Figure R1. Seasonal (left) and interannual variations (right) in the spatial extension of the three HNLC regions. Variations are referred to the mean extension of each region. The blue dashed lines indicate the regime shift occurring after 2010.

The EEP is a peculiar region that integrates subregions with 6-month out-phased seasonal variations. To better understand its variability we have split the analysis of the EEP into North-Equator and South-Equator regions. In Figure R1 we show that the southern equatorial region contributes more significantly to the mean extent of the EEP whereas the Northern subregion determines a large part of the observed variability (see also Fig R2 and R3). This suggests that the Equatorial signal is dominated by the large variability shown in the northern equatorial region.

[Figure]

*Fig R2. Time evolution of the total number of pixels covering each area: Whole equatorial region (in red). Pixels in the equatorial region which is located in the northern hemisphere are shown in black and those in the southern hemisphere in green. Note that while the mean value in the Southern hemisphere is larger, the northern region is more variable.*

[Figure]

*Fig R3. Time evolution of the total number of pixels covering each area scaled to the same range of values for better comparison:  Whole equatorial (in red). Pixels in the equatorial region which is located in the northern hemisphere are shown in black and those in the southern hemisphere in green.*

In Figure 7 we see an abrupt decline in the MOC around 2010 and then recovery to a level around 17 Sv which is both lower and more stable than before 2010. The authors seem to attribute a great deal of influence on oceanographic processes in all of the HNLC regions to this apparent "regime shift" (e.g., 404-406, 412-414). But given all of the higher frequency variability that is present in both periods, are the means for before and after 2010 even significantly different?

According to Moat et al. (2020), who suggest a previously unsuspected role for the AMOC in climate variability, the low AMOC event in 2009-10 coincided with a cold winter in Europe. MOI presents seasonal variability and it can be decomposed into a (1) seasonal, (2) irregular, and (3) interannual trend using different methods. We have used census-x11 method to identify each component (see R4). As shown in the figure below, the interannual signal for 2004-2009 yielded a mean value of 18.5±1 Sv, whereas the mean for 2010-2018 was 16.6±1 Sv. During the anomaly (2009-

2010), a mean value of 14.4 ±1.7 Sv was obtained. A detailed analysis of this change in meridional transport including a change-point analysis can be found in Moat et al. (2020). They report a change in the trend between 2008 and 2010, depending on the criterion selected (Fig. R5).

[Figure]

Fig. R4 Decomoposition of MOI into (1) seasonal, (2) irregular, and (3) interannual trends.

[Figure]

Figure R5. Change-point analysis of the AMOC–Ekman time series. (Moat et al. 2020).

Moat, B. I., Smeed, D. A., Frajka-Williams, E., Desbruyères, D. G., Beaulieu, C., Johns, W. E., Rayner, D., Sanchez-Franks, A., Baringer, M. O., Volkov, D., Jackson, L. C., and Bryden, H. L.: Pending recovery in the strength of the meridional overturning circulation at 26° N, Ocean Sci., 16, 863–874, https://doi.org/10.5194/os-16-863-2020, 2020.

The seasonal and interannual variability of the area are scales of variability that cannot be intercompared, or compared to the high-frequency signal. They represent different signals of the time variability of the HNLC area extensions. The statistical deviations of the mean area variability values have been shown as error bars in

Figure R1. This allows us to better compare how the mean variability statistically spread from the mean value (at the same time scales).

This seems like a question one would wish to ask before attributing far-reaching effects to this rather modest decline. Here again the mixing up of presentation of the results with discussion and speculation undermines the credibility of some of the claims made. I am particularly skeptical regarding claims that the MOC affects the extent of the SNP HNLC (406), and to a lesser degree the EEP one, and find the discussion of the underlying mechanisms to be quite speculative.

As we mentioned before, we have segregated results from the discussion in this new version. We agree that there is a degree of speculation in our discussion since the analysis is based on observations and we cannot identify mechanistic relationships. We can just suggest plausible explanations for the observed variability. Nevertheless, our data clearly shows a change in 2010 in the extension of HNLC regions that is in agreement with an event of low AMOC transport suggesting an influence of meridional overturning on the productivity of HNLC regions(see figure below).

A dynamically changing meridional circulation has significant implications for equatorial upwelling since, among other effects, influences the supply of nutrients to the biologically productive surface layer of the ocean. Meridional overturning depends, to a large degree, on deep-water formation, which, occurs in the high latitudes of the North Atlantic basin but not in the North Pacific. The most fundamental distinctions are (1) that the SNP does not ventilate the deep ocean at significant rates (Warren et al., 1983) and (2) that PMOC cell at this latitude corresponds to a rather independently functioning intermediate water cell. Indeed, according to Sigman et al. (2021), the SNP possesses an analog to the Southern Ocean's "upper cell" but lacks a clear analog to the Southern Ocean's "lower cell." Also, because is a shallower process, is more influenced by wind-driven processes and, therefore, presents greater interannual variability. It seems straightforward the influence of AMOC in the SO since waters are upwelled in this region. As in the case of AMOC, most subducted North Pacific deep water also upwells in the Southern Ocean (Thomas et al., 2021) and, therefore,  a similar effect is expected in the SO. I n the case of EEP, variability may depend on factors such as the strength of shallow overturning cells near the equator in the Indian–Pacific basin and on the inter-connectivity of the PMOC water with the rest of the global ocean. Also, the seaways in the Pacific are more complex, and determining the overturning signature is more challenging. Some studies have suggested that temperature anomalies subducted into the pycnocline at subtropical latitudes may not reach the Equator with any appreciable amplitude (Schneider et al., 1999a). However, mass water balances in the equatorial pacific reveal that the strength of the equatorial upwelling is related to variations in the PMOC (McPhaden and Zhang, 2002) and, therefore, an influence on EEP extent is expected.

In the case of SNP, the processes by which high-nutrient waters are maintained in the subarctic Pacific surface are not understood due to a lack of knowledge of the whole and detailed mechanisms by which nutrients return to the surface layer (Nishioka et al. 20201). The SNP lacks the deep circumpolar channel that characterizes the Southern Ocean and that allows the Ekman upwelling to draw large quantities of deep, dense, nutrient-rich water to the surface.  Rather, the meridional overturning cell subsystem of the North Pacific Intermediate Water seems mostly unconnected from deeper overturning cells (Talley 2013). The nutrient richness of the SNP upper water column appears to depend partly on diffusion-driven upwelling at the base of the pycnocline, which is enhanced by turbulence near steep bathymetric features (Nishioka et al., 2020). As we mentioned, MOC in this region is reportedly weak (1-4 SV) and extends no further than 50oN (Fujio et al., 1992; Ishizaki, 1994; Yaremchuk, 2001). However, there is evidence showing the response of this region to changes in PMOC (Burls et al, 2017). For example, it has been observed in TOPEX altimeter data that MOC influences the basin-scale baroclinic circulation in the SNP ( Kuragano & Kamachi, 2004).

In conclusion, the global overturning pathways for the well-ventilated North Atlantic Deep Water and Antarctic Bottom Water and the diffusively formed Indian Deep Water and Pacific Deep Water are intertwined (Talley, 2013). Accordingly, our results suggest a global scenario in which HNLC regions are susceptible to interbasin teleconnections rather than to local forcings. These general patterns can be modulated by feedbacks between different forcings. For example, PMOC variability and El Niño–Southern Oscillation are known to positively correlate (Tandon et al. 2020). In any case, we still need a clear understanding of the biological responses to these climate scale interactions in HNLC waters.

Fujio, S., T. Kadowaki and N. Imasato (1992): World ocean circulation diagnostically derived from hydrographic and wind stress fields, 1. The velocity field. J. Geophys. Res., 97, 11163–11176.
Ishizaki, H. (1994): A simulation of the abyssal circulation in the North Pacific Ocean. Part II: Theoretical Rationale. J. Phys. Oceanogr., 24, 1941–1954

Yaremchuk, M. I. (2001): A reconstruction of large-scale circulation in the Pacific Ocean north of 10°N. J. Geophys. Res., 106, 2331–2344.

 Kuragano & Kamachi, 2004Balance of Volume Transports between Horizontal Circulation and Meridional Overturn in the North Pacific Subarctic RegionJournal of Oceanography, Vol. 60, pp. 439 to 451, 2004

Thomas, M. D., Fedorov, A. V., Burls, N. J., & Liu, W. (2021). Oceanic pathways of an active Pacific meridional overturning circulation (PMOC). Geophysical Research Letters, 48, e2020GL091935. https://doi.org/10.1029/2020GL091935

McPhaden, M., Zhang, D. Slowdown of the meridional overturning circulation in the upper Pacific Ocean. Nature 415, 603–608 (2002). https://doi.org/10.1038/415603a

Trenberth, K. E., & Hurrell, J. W. (1994). Decadal atmosphere-ocean variations in the Pacific. Climate Dynamics, 9, 303-319. doi:10.1007/BF00204745

Schneider N., Miller, A. J., Alexander, A. & Deser, C. Subduction of decadal North Pacific temperature anomalies: Observations and dynamics. J. Phys. Oceanogr. 29, 1056–1070 (1999).

Warren B. A., Why is no deep water formed in the North Pacific? J. Mar. Res. 41, 327–347 (1983)

Talley LD (2013) Closure of the global overturning circulation through the Indian, Pacifc, and Southern Oceans. Oceanogr 26(1):80–97.

The "three major climate variability signals" (20) or "three main forcings" (95) of SST, ENSO, and MOC seems like a list that combines apples and oranges. In global mean SST, the biggest component of variability beyond the annual cycle is ENSO. So why does SST need to be included in this list? Anyway it appears that mean SST as an independent variable controlling HNLC extent is never actually discussed in the paper anyway; the cross-wavelet coherence results in Figure 7 are for ENSO and MOC. So why is it given such a prominent place in the Abstract and Introduction? This could confuse the reader about what their overall purpose is. (It is also probably an exaggeration to state that they "quantify the ... dynamic relationship between the observed Chl variability and three main forcings" (95). They do quantify the cross-wavelet coherence of NO3/Chl with the climate indices, but the discussion of the underlying physical processes is quite speculative.)

We recognize that the reviewer has a point here. There is no rigorous need to include SST and, in any case, it is not analyzed or discussed. We have removed the SST analysis.

The model-data comparison for NO3 could be expanded on a bit, e.g., "we found good agreement between nitrate in situ data and model results (r2=0.98)" (123). I think there should be Supplemental figures or tables that show the space/time domain of these comparisons, and break them down a bit more by region. To reproduce the gross spatial pattern of surface NO3, or especially the surface-to-deep gradient, is a very weak test of model skill. If one throws together data from all depths and from HNLC and nutrient-depleted subtropical waters, of course, you will get a strong correlation. If one looked only at e.g., surface concentrations in the SNP, one would get a very different result. How about including a Supplemental table that shows the correlation coefficients for the three major HNLC regions, for surface concentrations only?

Correlation coefficients between measured and modeled data for SO, SNP, and EEP are 0.74, 0.74, and 0.77 respectively. Nutrient concentrations in the SO are overestimated in 7.2 mmol m-3, and this has been corrected in our data. This is mentioned in the reviewed manuscript. The average profiles for each region are shown below.

[Figure]

Fig. R3. Mean global nitrate concentration and vertical profiles for each region.

Some details (Note that I have not listed here the numerous passages of Discussion that are vague or excessively speculative, but the authors should take note that there are many of these and try to trim them down (or shore them up with detail) as they restructure the paper overall.)

"and, therefore, of the withdrawal of atmospheric CO2" Withdrawal by what process on what time scale? HNLC regions per se do not affect atmospheric CO2, unless their extent is altered by changes in external supply of iron as suggested by Martin 1990 (see also 329)

We agree that the sentence was not clear. HNLC extent influences the potential co2 withdrawal since an increase in their global extent would result in a less productive ocean. Now it reads: 'their extent influences the potential withdrawal of atmospheric CO2 to the deep ocean'

"oligoelements" unnecessary jargon. Changed by 'elements'

"coarsely known aspects" not clear what this means. The sentence has been rephrased to 'Only general aspects such as expected shifts in phytoplankton community composition or changes in Fe-cycling rates have been addressed to date (Fu et al., 2016; Lauderdale et al., 2020).

"it is arguable if these ephemeral systems share structural and functioning similitudes with the large HNLC regions" it is uncertain whether these ephemeral systems share structural and functional similarities with the large HNLC regions

Thank you. Corrected

"reporting a positive North Pacific Gyre Oscillation (NPGO) and nutrient correlation" reporting that surface nutrient concentration was correlated with the North Pacific Gyre Oscillation (NPGO)

Corrected change "nutrient outputs" to "nutrient concentrations"

Corrected

"a good indicator to describe the value overall phytoplankton trend" a good indicator of the magnitude of the overall phytoplankton trend?

Corrected change "climatological indices" to "climate indices"

Corrected change "the pauperized subtropical gyres" to "low-latitude oceanic waters"

changed

220-224 I think this discussion neglects Eastern Boundary Currents, which represent one of the largest areas of consistently high Chl + high NO3

Yes, we mention in the introduction these regions. Eastern boundary currents are not HNLC regions and therefore we do not discuss them.

delete "i.e."

deleted delete "values"

deleted

"has remained elusive since ... requires coherent information" something missing here

The sentence has been rephrased. Now it reads: Systematically determining the boundaries of HNLC regions has remained elusive as it requires coherent information from both nutrient and Chl fields

I actually think Figure S2 could be in the main text. It would help the reader to understand what the authors are doing.

Figure S2 has been included now as Fig.1

change "corresponding" to "correspond"

changed change "therein" to "there"

changed add a comma after "ratios"

added change "ice sheet" to "sea ice"

changed

"exhibits a differentiated dynamic" I can't tell what this means.

Changed by: exhibits distinct Chl variability patterns.

"phenological variations" I don't think this term is useful or necessary here

It now reads: but these variations

"This region is also subjected to zonal variations" How about "This region has distinctive eastern and western regions"?

Changed by: This region exhibits distinctive eastern and western provinces

"in which ocean productivity ... importance of advection of Fe" something missing here

It now reads: is a more enclosed basin in which ocean productivity is driven by the advection of Fe

"trend robustness is provided by the coherence in the time series obtained using SOM" I can't tell what this means

It now reads:  In our case, robustness in trend analysis is provided by the spatial coherence in the time series obtained using SOM classification. This methodology is based on the similarities in the temporal variability patterns and it clusters regions with similar trends and variability.

"not exclusive of oceanic Fe-limited waters, since it has been also observed in the highly productive Patagonian shelf" Not clear what they are trying to say here. The Patagonian shelf is not oceanic and is not Fe-limited.

Correct. This is our point. We report a shift that it is also observed in other southern regions (not only in HNLC regions)

add ", respectively" after "100% in April and 70% in July"

Corrected

"The extension of the HNLC region in the boreal winter is the boreal winter is 25%" ???

Some words were missing. Now it reads: Indeed, the extent of the HNLC region in the boreal winter is 25% lower than the mean annual extent.

change "thought" to "though"

corrected

" As shown in Figures 6 a and b, the temporal variability of both the characteristic NO3:Chl ratios and SST at each region peaks at 12-month periodicity, being this seasonal modulation more intense and temporally consistent in the case of temperature at high latitudes and weaker in the equator." This is very poor scientific writing. The result being presented is rather mundane: the most obvious detectable periodicity is the annual cycle, and seasonality is stronger at higher latitudes. Please rewrite.

References to SST have been deleted, following the revierwers' recommendations

"transference from annual to semiannual periods since 2010" Not sure what the right word is here but I am fairly sure "transference" is not it. How about "display a semiannual mode, which accounts for a larger fraction of variance after 2010"?

Thank you. We have rephrased the sentence change "in" to "on"

changed delete "value"

changed change "phytoplankton uptake" to "phytoplankton biomass"

changed

"Semiannual cycles" I try to avoid referring to variability at periods other than annual as "cycles" (excepting Milankovitch frequencies of course, but this paper is concerned with subannual to decadal scales) (see also 355, 433)

While we believe that conceptually is not incorrect, we agree that the use of cycles is arguable when a variation is not markedly repetitive, like tidal cycles. We have rephrased the sentence to: Semiannual variability generally occurs in regions where warming and cooling phases show different durations.

change "Contrastingly" to "Conversely" or "By contrast"

Changed change "phase out" to "out of phase"

Changed

"suggesting a meridional propagation of the MOC effect" vague

The sentence has been rephrased add a comma after "AMOC"

Added change "more unclear" to "less clear"

Changed sea ice or glacier ice?

Riebesel et al 2009 mention sea ice delete ", also based on remotely sensed Chl,"

deleted change "this effect is unlikely" to "this effect is unlikely to be important at the time scales considered here"

Changed

"retrieved from the increasingly improved and longer and longer time series of remote sensing observations" retrieved from time series of remote sensing observations of increasing duration and quality changed delete "through complex processes"

deleted

Figure 1 - the white contour lines are difficult to see in some places

Contourlines have been changed to black.

Figure 2 - the red lines that indicate linear trends don't look like straight lines to me, but it's hard to tell

Red lines (now blue) are straight. However, we acknowledge that because of the image resolution they appeared somewhat stepwise. We have increased image resolution and the colors have been switched to improve the visibility of the line.

Figure 5 - what exactly the y axis represents is not clearly explained; the meaning of the different colored bars is fairly obvious but should still be stated

We have modified the legend. Now it reads: Figure 6. Seasonal (left) and interannual variations (right) in the spatial extent of the three HNLC regions, are represented as a percentage of variation from the mean extent of each region. Dark and light coloured bars indicated positive and negative values, respectively. The blue dashed lines indicate the regime shift occurring after 2010.

English/formatting

One quirk of English usage that appears over and over is using the word "extension" instead of "extent". There are 27 in total and I think "extent" is more appropriate in virtually every case. Another is using "at" in place of "in" a region, e.g., "at SO", "at the EEP". I would write "in the XXX" in all cases. (Interestingly, it used to be fairly common to use "at" wrt cities, as in "I attended the AGU meeting at San Francisco". But this fell out of common use a long time ago.)

We apologize for the misuse of the noun extension ('expansion') instead of 'extent' which refers to a range of locations, being more appropriate in this case. It has been corrected all through the document. As non-native speakers, we also appreciate clarification about the use of 'at'. This has also been corrected.

Numerous references are missing from the reference list, e.g., Garnesson/Grarnesson et al., 2019 (spelling varies); Green et al., 2017; Ibanhez et al., 2017; Kumar et al., 1995; Martínez-García et al., 2009; Qui, 2002 (probably Qiu). This is NOT an exhaustive list. I have doubts about whether Takeda 2011 is a traceable reference (searching on the doi turned up only stale links).The reference format is inconsistent in the sense that multiple references within a parenthesis are sometimes arranged alphabetically, sometimes chronologically.

All references have been reviewed and arranged chronologically in the parenthesis. Regarding Takeda, you are right. The paper is freely available from several sources (researchgate, semanticscholar) but the doi link does not work. We have removed this link.

---

## Author Comment (AC3)

**Reply comments Referee #2:**

This paper relies on the spatio-temporal variability of the HNLC regions identified by SOM over the three major ocean areas using satellite-derived chlorophyll-a and modeling outputs of nutrients. The authors have performed the NO3:Chl as an indicator of the distribution limit of HNLC. They have demonstrated the linkages between HNLC extent and some climate-driven factors and teleconnections.

As a first very general comment, I would say that this is a valuable case study that can be published with some major corrections. The authors presented a lot of data and analysis procedures that need accurate processing schemas and precise interpretation, which they handled well.

The introduction section is well written and presents an adequate understanding of the presented work. The methodologies are adequate, but need some improvements. Some supplementary materials may be inserted into the main text because of their essential investigations and frequent references to them (e.g., Fig. S2).

We are grateful for the generally positive review of RC2. We have improved some aspects of the M&M section and a figure from supplementary material is now included in the main text as Fig.1

The Spatio-temporal variability of HNCL/NO3:Chl should be presented more precisely and quantitatively. For example, maps of monthly climatology, charts and maps of inter-annual cycles, and spatial-temporal cross section such as Hovemoler chart may present valuable results.

The aim of the SOM analysis is precisely to replace this type of traditional climatology analysis with more objective, quantitative, and precise approaches. SOM analysis allows the reconstruction of the spatial patterns and their temporal dynamics (seasonal and interannual) of the study variables (i.e. using the results shown in Figures 4, 5, and 6). The SOM mapping has advantages over other methodologies, e.g. monthly climatology, maps of inter-annual cycles), as relevant time and spatial scales are unveiled without any prefixed assumption. For instance, if two different spatial patterns occur in the same month with the same probability, using a monthly average only one spatial pattern will be obtained, while SOM should identify both patterns. We explain this more profusely now.

Some of the findings of the results section have not been well demonstrated in this study (e.g., section 3.4.1 Influence of SST variations – the global power spectra of CWT are needed to explain the intra-annual and inter-annual cycles). Some contents of Figures need to be better framed/explained. Some words would be expressed in accurate form. e.g., it is not clear in many places in the text that the "wind" word is mean wind speed, or wind vectors, or wind stress, or wind components (zonal and meridional).

We have removed the section explaining variations in SST because it is not a forcing. The term wind has been reviewed throughout the document and, particularly in section 4.2, and we specify now whether we refer to wind intensity, etc..

I think a discussion section is required to explain the performance and limitations of the presented methodologies and results, which are not seen in this manuscript.

Following the reviewer's suggestion, we have reformatted the results and discussion section into separate sections and we further discuss the presented methodologies.

Bioregionalization analysis was used in previous studies to classify the global oceans (e.g., Longhurst, A. (2007), Ecological Geography of the Sea, Academic Press, London). Could the author highlight the need to new ocean's regionalization which are not accessible from available global regionalization? And why SOM and not the other classification methods such as k-means?

Bioregionalizations can be based on different variables and generally differ in that they obey different requirements. It is simply not possible to analyze certain environmental issues from regionalizations based on low-resolution information or criteria that do not respond to the object of the study. This is particularly true in the case of the study of pelagic organisms where the biome extent changes over time. The first regionalizations proposed by Alan Longhurst (1995 and 1998) obeyed geographic divisions based on the average distribution of phytoplankton biomass and primary production and provided a coarse but useful division of the global ocean. In subsequent analyses, other components of the trophic chains were incorporated. In specific cases such as HNLC, the biome presents highly dynamic boundaries and it becomes necessary to resort to techniques that incorporate this variability.

Self-Organizing Maps (SOM) and k-means are both unsupervised machine learning algorithms, but they have different characteristics and are used for different purposes. K-means is a clustering algorithm that groups similar data points together based on spherical clusters, while SOM finds the best matching unit (BMU) for each data point, which is the node in the grid that is most similar to the data point. Also, SOMs can handle non-linearly separable data.

One of the advantages of the SOM over k-means is that it performs the groupings based on the shape of the time series, which provides very coherent regions with similar dynamics, which allows each subregion to be characterized later based on robust statistics.

The SOM non-linear mapping has also advantages over linear methodologies, like principal component analysis, empirical orthogonal functions (EOF), and even K-means. If the data distribution on a two-dimensional space has a correlation close to zero, this can be difficult to find it using PCA or similar, however, by using SOM, the resulting weights will be adjusted in such a way as to match the shape of the data distribution. K-means is a special case of SOPM in which the neighborhood function is zero (not considered).

I think the findings of the nutrient model are not presented well. Some additional information is required.

We do not run a nutrient model. Results are from the numerical simulation PISCES produced at Mercator-Ocean (https://www.pisces-community.org/ https://www.pisces-community.org/index.php/model-description/). It runs over NEMO hydrodynamic model, a primitive equations model https://www.nemo-ocean.eu/ and therefore, it does consider MLD. However, we did compare model results with available data globally and regionally and we provide now the correlations between model data and regional data for global and regional data. We also include now in the supplementary information the mean NO3 profile in each region depicted from in situ data,

The Mixed Layer Depth (MLD) is one of the main oceanographic indicators that can be used for the interpretation of nutrients and phytoplankton variabilities. Does the nutrient

model consider the MLD? If yes, please indicate in the text. And if not, I think it is required to consider the global ocean MLD in your work.

Yes, it considers the MLD. Answered above

Looking to be constructive, in addition to the overall comments above, which should be taken into account in a possible review, I would like to point out the following Remarks.

1- In the 2.1 Ocean color data:

The GlobColour data are presented in 25 km spatial resolution globally. The authors have mentioned that the composites have a 0.25° spatial resolution. We know that the spatial resolution of 25 km and 0.25° are not the same specially at higher latitudes. If the 0.25° is true, please explain the spatial interpolation methods. If not, please correct.

The reviewer makes a good point here. The GlobColour level-3 mapped products have a resolution of 1/24°, 0.25° or 1.0° (i.e. respectively around 4.63 km, 28 km, and 111 km at the equator) for global products. They consist of the flux-conserving resampling of the global level-3 binned products. The geographical location and extent of each bin is determined by the so-called Integerized Sinusoidal (ISIN) grid. In particular, we use data from the rectangular regular map product provided by GOBCOLOUR at global scale, with a regular grid in degrees, with a spatial resolution of 0.25 degrees, 27.82km at the equator that varies with the latitude. The spatial extent was given in pixels and it has been changed to 'real' areas in km2 in this new version. Data are not interpolated since they are structured in a regular grid.

As a result of these changes, small variations are observed in the mean climatology of the area anomaly for the three HNLC regions. Intra and interannual area variations in EEP and SNP regions are slightly larger when computing the area based on the linear dimension (i.e. km²) than on the number of pixels.

[Figure]

[Figure]

Figure R1. Seasonal (left) and interannual variations (right) in the spatial extension of the three HNLC regions. Variations are referred to the mean extension of each region. The blue dashed lines indicate the regime shift occurring after 2010.

2- In the section: 2.2 Nitrate data the authors have made some essential assumptions that need to be approved precisely. May be explain more in the Results.

We now explain better matching between model and in situ values in each HNLC region. We also include nutrient profiles in the supplementary information.

3- Please provide more information about the setup of the SOM algorithm (which are available not in the text nor supplementary material), in particular about the initial configuration: linear or random initialization, sheet or toroidal network, etc. Which neighbor function (Gaussian, Ep, et) is used to update the neighbors of the excited neuron (BMU) after each iteration during the training process.? Did the authors check the sensitivity of the SOM pattern to linear and random initialization?

The SOM algorithm is composed of two main steps: initialization and training. In the initialization, the architecture of the neural network used in this study is set in a sheet hexagonal map lattice of neurons, or units, in order to have equidistant neurons (to avoid anisotropy artifacts). Each unit is represented by a weight vector with a number of components equal to the dimension of the input data, i. e. number of rows or number of columns in the Chl and NO3 matrices, depending on whether the analysis is performed in the temporal or in the spatial domain. We use an initial network composed of units of random values (random initialization). In the training process, the initial neural network is transformed by iteratively presenting the input data. In each successive iteration the neuron, or unit, with the greatest similarity (excited neuron), called Best Matching Unit (BMU) is updated by replacing their values with the Chl and NO3 values of the input sample data. The similarity is estimated by computing the Euclidean distance between the components of the input sample and the components of the weight vector of the unit. The unit most similar to the input sample is the one with the minimum distance. In the learning process, Chl and NO3 values of the neighboring neurons of the excited neuron are also updated replacing their values with values determined by a neighborhood function. In this way, the topological neighbors of the BMU are also updated through the neighborhood function. In this study, we use the imputation batch training algorithm (Vatanen et al., 2015) and a Gaussian neighboring function. We do not find important differences between linear and random initialization. However, linear initialization is more computationally expensive.

4- The output of the SOM is also not well defined beyond being a map or topology; for example, how are the errors computed? The number of neurons is chosen not only depending on the topological errors (or topographic errors), but also on quantization errors. How different are the patterns when the number of neurons is, for example, 9?

The size of the neural network (number of neurons) depends on the number of samples and on the complexity of the patterns and an optimal choice is important to maximize the quality of the SOM. In the present study, the map size is set to be [4 x 3] with 12 neurons for the time domain analysis, and a [3 x 3] neural network is used in the spatial domain. Using larger map sizes, the patterns are slightly more detailed, and more regions of a particular variability emerge, but the occurrence of the probability of the patterns decreases, without affecting the results noticeably (Basterretxea et al., 2018; Hernandez-Carrasco and Orfila, 2018). If a reduced neural map, such as [2 x 2] is used, patterns are concentrated together with the occurrence probability in a few rough patterns but increasing, in this case, the topological error.

5- Figure 6 and 7. The arrows are too small.

This has been corrected.

6- I recommend to show the time-series anomaly of HNLC and teleconnections at the top of the Fig. 6 and indicate the significant inter-annual cycles.

Figure 6 has been removed following a reviewers suggestion

7- It is hard for readers to infer the shift after 2010 (Fig. 5). I think more visualization/explanations of data are required.

We indicate the shift with a blue dashed line in the figure.

8- There are some abruptions in the significant annual cycles (1.5 year, from year 2006 to 2010) in the Fig. 7 which need to be explained. I suggest perform the global spectral cycles graphs.

We have improved the figure including mean spectral peaks and further explanation is provided.

9- The authors have considered SST and the teleconnections as a factor controlling HNLC variability. Are there any environmental factors such as precipitation and wind stress that may affect HNLC variability? Please discuss.

We have segregated results from discussion and a more detailed discussion on teleconnections is now provided. This includes ENSO associated effects. However, the analysis of particular factors, such as rainfall, is beyond the scope of the present study.

---

## Author Response (AR2)

Dear Editor,

Please find attached a revised version of our manuscript egusphere-2022-827 entitled "Global variability of high nutrient low chlorophyll regions using neural networks and wavelet coherence analysis" which we have modified according to the suggestions and criticisms of the Referee. We also provided a detailed letter that addresses point-by-point all reviewers' comments.

We acknowledge the Reviewers' constructive comments which helped us to improve significantly the discussion of the results regarding the impact of the large-scale forcings on the variability of the NO3:Chl ratios. We also thank them for the overall positive appreciation of our work. We sincerely hope that, with the careful revision we have made, our paper will be found suitable for publication in Ocean Science.

**Detailed response letter:**

This paper has improved markedly in the revision. I congratulate the authors on having made a serious effort to address the previous critiques. I have a few further (mostly minor) comments below. (Note that the page numbering scheme is odd: it is not sequential but it doesn't reset to 1 on each page either; possibly it is sequential but the first digit got cut off for #'s >100. In any case, when I refer to X/Y I mean page # / line # and there should be no ambiguity.)

**Major points:**

(1)      I think these authors have done a very good job of separating the Discussion from the Results. There are one or two passages of what I would call Discussion in the Results (e.g., 10/00-02, 11/45-48), and some parts of the Discussion are vague or too speculative. But generally, they did an excellent job.

We appreciate the recognition from the reviewer. We have modified/or eliminated these paragraphs to be more assertive and robust in our arguments. For example, the sentence in 10/00-02 was unnecessary and it has been deleted.

(2)      There are a few places where "facts not in evidence" are referenced, or prematurely introduced before the data are shown. For example, the step change after 2010 shown in Figure 6: I think it would be better to just say "see next section" when this topic is introduced on 11/33, rather than referring the reader to Figure 3, because this step function is difficult to discern in Figure 3. This topic is discussed again on 11/54-55, and another ambiguous data reference is introduced, "i.e., P3 substitutes P1". This may be true but it's hard to tell from the plot: it is obvious that P1 disappears but less obvious that P3 becomes dominant in the months when P1 previously had been. It might be better to just say "P1 is no longer observed".

We recognize that some aspects regarding the step change in 2010 are scattered throughout the results. They recurrently show up in the different analyses carried out. We have modified P 11/33 and P11/54-55 as suggested.

(3)      Parts of the Discussion are excessively speculative, or appear to contradict the previous text, and I have doubts about whether some of the literature cited is interpreted correctly. I commend the authors for their thoroughness; they cite quite a few references I had not heard of. But some that I am familiar with, including classics like Gargett 1991, are not necessarily interpreted correctly. The large-scale net Ekman transport is upward, albeit sluggish. I'm not saying localized upwelling associated with either atmospheric or oceanic

cyclones is unimportant, but that's not what Gargett (1991) is about. In the case of Cullen (1991) (14/47) I also have difficulty connecting the assertion made to the contents of the cited reference. The assertion that " that generate enough Ekman pumping to maintain high nutrient concentrations in the near-surface waters (Gargett, 1991; Harrison et al., 2004)" is questionable and likely to be misinterpreted. The Alaska Gyre IS a cyclone (whereas the NPSG is an anticyclone).

Gargett 1991 is cited because, while certainly, it is not focused on cyclones, it refers to the necessity of strong wind forcing required to debilitate the strong density stratification hampering vertical nutrient delivery in the North Pacific. We acknowledge that his analysis extends beyond the effects of wind-forcing and it could be misinterpreted, therefore, we have removed the sentence. Besides this paragraph, we have removed other sentences which could be considered too speculative.

Boyce et al 2010 drew some rather vigorous criticism (www.nature.com/articles/nature09953). Boyce et al (2014, 10.1016/j.pocean.2014.01.004) address some of these criticisms and could be cited here. They claim that the basic conclusion that a long term secular (downward) trend is detectable remains sound, but this conclusion remains controversial and I think that the authors of the current contribution should treat it skeptically.

We agree with the reviewers' view on Boyce 2010. We have substituted this reference in the introduction by the 2014 publication, which is more robust, and rephrased the sentence on chlorophyll trends to avoid being conclusive.

The results of Polovina et al (2008) represent too short a time series to be inferred to represent a long-term secular trend or anthropogenic warming signal.

We agree it is a short record (9 years), but they are published results that, we believe, should be cited in the introduction as an example of previous studies. Anyhow, we have toned down the relevance of this paper.

The entire paragraph on 14/53-60 contains a number of questionable assertions. The source of dissolved Fe in the northwestern Pacific is primarily from shelf and slope sediments, not river discharge, and the primary source of vertical mixing is from tidal currents (cf. Nishioka).

We do not think that the paragraph is questionable, yet we recognize that it provides a rather simplistic explanation. We have expanded our argument based on the review by Nishioka et al (2021) who in detail discuss the sources of iron and nutrient supply in the subarctic Pacific. We agree that most references indicate that shelf sediments are a main source; however, some authors (ie. Nagao et al 2007), cite both atmospheric and river discharges along the western coast may be important.  Indeed, atmospheric dust has been considered to be the most important source of Fe in the North Pacific affecting biological production (Uematsu et al. 1983, Duce and Tindale 1991 and Mahowald et al. 2005, Boyd et al. 1998) though, more recently, Boyd et al. (2010), questioned dust-mediated phytoplankton blooms.
Nishioka (2021) proposes a coherent explanation to conciliate the putative sources for the biological response in the SNP that incorporates knowledge of both the atmospheric and oceanic Fe supplies. Blooms in open waters would be controlled by the sedimentary Fe supplied from the continental margin circulated laterally through the intermediate layer and upwelled to the surface by several mixing processes (winter mixing and eddy diffusive mixing), including interactions of tidal currents with the rough topography, whereas sporadic and

patchy surface phytoplankton production observed in the absence of vertical mixing in spring and summer is attributed to the atmospheric input of Fe dust (Nishioka et al., 2021).
The section on 16/97-03 is also confusing. I agree that the extent of the EEP HNLC is primarily a function of ocean upwelling rather than aeolian dust deposition. But what is stated here ("we observe a reduction of the extent of the HNLC region in the EEP during an enhanced wind intensity period") is actually the opposite of what is expected, and what is shown in Figure 6. In the EEP the period of strongest upwelling is in the boreal summer (e.g., Philander and Chao, 1991, JPO 21: 1399), which is associated with an expansion of the HNLC in Figure 6. The same should be true on interannual time scales: a period of increased trade wind intensity should be associated with an expanded HNLC, not a reduction (see also 17/44-45).

We agree that that sentence is confusing and contradictory with the pattern observed in seasonal variations (Fig 6b). This has been clarified in the revised version of the ms. Nevertheless, the extent of the HNLC regions is not necessarily solely dictated by the strength of the upwelling, but it is also defined by the interaction with adjacent circulation patterns (i.e. subtropical gyres in this case). This has been observed in coastal systems where upwelling intensification is correlated with water temperature but not with surface extent. In the case of HNLC regions, upwelling strength is likely to control excess nitrate concentrations but the extent obeys to more complex equilibria.

The reference to a "meridional propagation of the MOC effect" on 17/33 also seems backward to me: if the fluctuations in HNLC area arise from fluctuations in the strength of the MOC, should they not appear first in the SO and last in SNP? And then on 17/42 we have "weaker MOC is related to increases in the extent of the EEP and contraction of the SNP and the SO HNLC regions". This appears to contradict what was said above about stronger MOC implying stronger SO upwelling (17/29). Maybe that's just me making a simplistic assumption that stronger upwelling leads to an expanded HNLC. But maybe this is not the case in the SO, and the data do seem to show that the SO HNLC contracts after 2010. Possibly these things are related: the reason there is a lag between the EEP and the SO is that the changes in the SO HNLC extent are only affected by the declining phase of MOC fluctuations, not the strengthening phase. Why this would be I do not know, but it may have something to do with nutrient stoichiometries in the upwelled waters (e.g., 10.1038/nature02127; 10.1038/nature04883).

There are several aspects regarding this question. First, MOC influences upwelling rates and coupling appears to be more significant when it debilitates after 2010, as shown in Fig. 7f,g,h. Indeed, NO3 is reduced but CHL is enhanced, particularly in the declining phase of MOC. A plausible explanation for increased CHL values is that the plankton community can cope with Fe recycling rates under not strong upwelling conditions (i.e. weaker MOC), as reported by Rafter et al. 2017, and thus a better agreement. This occurs at decadal scales or, as a general pattern in the analyzed series.

Secondly, it should be also considered that, as we mentioned above, HNLC extension is not necessarily related to upwelling intensity. However, in the case of the shift in 2010, our analysis reveals a major change in the NO3:CHl patterns. Pattern P1 disappears in favor of P2 and P3 which show both less extension and reduced NO3:CHL ratios, especially in the SO (see Fig. 4 and 5a).

A more complex question is that regarding the phase relationship inferred from the wavelet coherence. Here the analysis reveals the coherence at monthly scales. During a weakened

MOC, there is an antiphase coupling in the SO at ~1 year time scale. This means that increases in MOC rapidly produce NO3:CHL increases in the SO, i.e. HNLC intensification. The pattern in the SNP is more complex because it is on antiphase. This is, MOC increases reduce NO3:chl ratios which would indicate that MOC intensification putatively affects Fe availability in this region, therefore, increasing ocean productivity. In general, our results reveal that, although significant after 2010, MOC variations produce short-time (<1 yr) responses in the three systems.

(4)    The assertions regarding model-data agreement for NO3 (e.g., 5/50) are still unconvincing. Again, if you look at concentrations over depths where a strong vertical gradient exists, you are always going to get a strong correlation. My previous suggestion to look at surface concentrations only was not followed.

We believe that this is not well explained in the M&M section. We are comparing 'mean 0-20 m' NO3 values. Average values are used because surface sampling depths vary between cruises. Multiple casts do not contain surface (0m) values. While a 0-20 m average may somewhat filter undetectable NO3 levels at the surface we do not think that it should highly influence overall correlation. Indeed, CMEMS has published a quality assessment document of their modeling products in which NO3 data are compared with measurements of Argo data with similar results (see Page 27/ 49, https://catalogue.marine.copernicus.eu/documents/QUID/CMEMS-GLO-QUID-001-029.pdf). 'Overall, the model and WOA-2013 climatologies are in good agreement. Exceptions can be pointed out in the Southern Ocean, where nitrate levels are too strong in the model, and along the Arabian Peninsula (Yemen and Oman) and in the Bay of Bengal where nitrates are significantly weaker in the model'. General correlation is similar to our analysis (0.98), however, they use the complete dataset (all depths).

(5) On p. 10 it is stated that "It is noteworthy that nutrient concentrations are generally lower in the SNP (i.e. <17 mmol m-3) than in the SO while biomass is comparatively higher (see Table 1)." This is true, but when I look at the Table, the near-constant value of [NO3] within each Rx cluster is quite remarkable. I interpret this as evidence for the robustness of the method, and I think this is something that could be commented on in the Discussion.

We appreciated this comment that has been included in the first paragraph of the discussion section.

(6) Is there any evidence for a "regime shift" in 2009-2010? I am wondering if any of the cited references discuss this.

There are multiple pieces of evidence in the literature of this event, including changes in MOC, variations in the Pacific sea level, and changes in chlorophyll. However, we have not found any in-depth analysis of the causes and long-term consequences of this episode.

**Some details:**
1/32 change "budgets" to "inventory"
Changed
2/35 Brindley misspelled (see also 25/26); Tyrrell misspelled
Corrected

2/48-49 change "iron is required in largest amounts than any of the trace metals" to "iron requirements are the largest among the trace metals"
Changed
2/53 change "nutrient" to "macronutrient"; delete "often"
Changed
3/70 change "biological structure" to "ecosystem structure"
Changed
3/73-82 could add some additional literature citations here e.g., 10.1038/s41598-018-37436-3 (note that the method appears to detect two patterns that are seasonally HNLC: see Figure 3)
Ref to Birchill et al has been included.
3/81-82 change "structural and functioning similitudes" to "structural and functional similarities"
Changed
3/97 change "in" to "on" and "response" to "responses"
Corrected
4/21 delete "that varies with the latitude"
Deleted
4/35-5/40 Something is wrong here: the text takes an abrupt zag from model description to climate indices and back again. Some text was accidentally spliced in the middle of another paragraph.
The reviewer is correct. In this version, for unknown reasons, the paragraph was spliced in the middle of another paragraph. We apologize.
9/81 "is not more exceptional" I can't really tell what this means
It should read 'is more exceptional'. Corrected
9/84 I would consider also citing 10.1038/nature07716 here
We have included th suggested reference (Pollard et al., 2009).
9/94 change "distribution" to "mode"
Changed
10/15 delete "coupled"
Deleted
10/16-17 I would replace "since" with a ; or a : and change "duplicates" to "doubles"
Changed as suggested
10/19 change "biomass" to "chlorophyll"
Changed
10/22 I would consider also citing Harrison et al 2004 here
Done
11/38 "Most differentiated patterns, also displaying the highest probability of occurrence (the probability to find a pattern similar to the input data)" The most differentiated patterns, also displaying the highest probability of occurrence (the probability of finding a pattern similar to the input data)
Corrected
11/41 "scenarios" seems like an odd choice of terms here
Changed by patterns
11/52 "winter" should be spring (April)?
Yes, it is April (spring)
11/64 the positive anomaly appears larger in April than in March
Corrected

12/65-66 "boreal winter" should be "austral winter"? 25% exceeds the limits of the graph which shows max deviation as <20%

Corrected

12/78 delete "the year"

Deleted

12/78-79 "when ENSO variability intensified" Is there a data or literature reference for this? I have not heard of this before.

There is a paper by Cai et al 2022 on ENSO variability. However, we refer to Fig 7b. The sentence has been rephrased.

12/92 delete "the analysis of"

Deleted

13/98 change "discriminate" to "delineate"

Changed.

13/04-05 mention the physical ocean as well here (e.g., upwelling)?

A mention to physical ocean procesess has been included

13/10 delete "fields"

Removed

13/12 change "complex" to "diverse"

Changed.

13/17 delete "unproductive"

Removed

13/17-18 "the SO is the largest region presenting clear latitudinal variation in the characteristic Chl patterns" This seems to conflate two separate points: the SO is the largest HNLC region and it is the ONLY one that shows clear latitudinal variation in the characteristic Chl patterns (Figure 3).

It has been rephrased to: 'the SO is the largest region and it is the only one presenting clear latitudinal variation in the characteristic Chl patterns'

13/21-22 change "Both the SNP and the EEP respectively constitute 8% of the total HNLC" to "The SNP and the EEP each constitute ~8% of the total HNLC"

Corrected.

14/30 2019 should be 2009?

Yes. Corrected.

14/31-33 "This non-linear enhancement in phytoplankton, which is not exclusive to oceanic Fe-limited waters (see for example Marrari et al., 2017), positively biases the Chl increase rate in these subregions." Meaning is not clear, and the entire sentence is probably expendable.

We agree. It was intended to be emphatical but it is not necessary.

14/61-62 "where seasonality is marginal" This does not appear to be the case in Figure 6.

We changed 'marginal' to 'weak'. As shown in Fig. 6, the maximum values of % variation at EEP are 8% while the other two regions display much higher variability (20% and 50%).

15/63 "subregions with 6-month out-phased seasonal variations (north and south of the equator)" data reference? where is this shown?

We do not think that a reference is required here since it is quiet straightforward (although may be not well expressed). Seasonality is 6-month out-phased in the northern and southern hemispheres (i.e winter in the N is summer in the S). The EEP extends over both hemispheres, and although the seasonal signal at these latitudes is weaker than at higher latitudes, it is not negligible.

15/71 "the seaways in the Pacific" circulation pathways?

Yes, corrected

15/78-79 "corresponds to a rather independently functioning intermediate water cell" rather corresponds to an independently functioning intermediate water cell

Changed as suggested

15/79-82 references to MOC here are ambiguous: if they are talking about the global MOC rather than PMOC it might be a good idea to put "global" before "MOC" for clarity (see also 15/87 "meridional circulation")

It refers to PMOC. Corrected

15/88 change "variations" to "variability"

Changed

16/06 "in this region" unclear antecedent; if still talking about the EEP here, please specify

Yes. It is specified now.

16/08 "the ENSO index" which index? there are many ENSO indices to choose from

This is detailed in M&M section, Pag 3. 'El Niño Southern Oscillation Index (MEI.v2), hereafter ENSO index'

16/12 delete "events"

Deleted

16/12-14 I think this whole sentence is expendable.

Deleted

16/20 change "ENSO-related equator-originated sea surface height anomalies" to "ENSO-related sea surface height anomalies originating in the tropics"

Changed as suggested

16/23 change "which nicely explains the fluctuations of salinity, nutrients, and chlorophyll" to "which explains strongly correlated fluctuations of salinity, nutrients, and chlorophyll"

Changed as suggested

17/34 "Figure 8b also reveals a decline of the MOC until 2010". This Figure does not appear to exist. (and change "until" to "around")

It should have read Fig. 7b (now 7e). Corrected

17/46 "the slowing down of the overturning circulation in the Pacific Ocean since the 1970s" This statement seems incongruous given that the cited reference is > 20 years old. Do we know that this trend continued or consolidated? Or is this a case of interdecadal variability that manifests as a slowdown over the period studied but has since reversed?

Detailed information on long-term PMOC variability and its influence on EEP biogeochemistry is scarce. We are aware that the data reported by McPhaden and Zhang, 2002 is old and probably reflects a subdecadal variability. Nevertheless, it provides evidence of the proposed relationship between PMOC and equatorial upwelling. We have rephrased the sentence to evidence that this is not a general trend.

17/52 "an atmosphere's energy balance indicator" I don't think this is necessary.

Removed

17/58 change "reduce" to "reduced"

Changed

18/63 change "is considered" to "occurs"

Changed

18/63 change "Paleoceanographical records reveal a strong correlation between proxies of aeolian Fe flux and productivity has been reported" to "Paleoceanographical records showing a strong correlation between proxies of aeolian Fe flux and productivity have been reported"

Corrected

18/65 "in present times, dust deposition in this area has notably varied" Is there a literature reference for this (e.g., 10.1073/pnas.0607657104)?

Reference to McConell has been included

18/76 "further evidence of the global scale coupling and feedback between the atmosphere, the ocean, and global productivity variations" Possibly this is true, but nothing shown in this paper necessarily depends on ocean-atmosphere feedbacks. Similarly with "anomalies in global forcing intensity". It's not clear what is meant by "forcing intensity" here, but I do not think anything shown here requires a change in e.g., global net forcing of climate by GHGs, or a change in the global mean surface ocean wind stress.

"further evidence of…' has been removed and "forcing intensity" has been changed by global ocean circulation patterns

191/4 URL for CARINA is outdated (should be ncei.noaa.gov)

Corrected

19/22 Aumont reference cites Discussion paper; should cite final published version

Corrected

20/50 Nojiri misspelled

Corrected

Reference format is still inconsistent in that journal titles are sometimes abbreviated, sometimes not; sometimes all words capitalized, sometimes not; author names sometimes spelled out, sometimes not;

The reference format has been corrected following the bibliographic style of the journal.

Table 2 caption should define "Max variation" and state monthly or annual mean data

It now reads 'Basic statistics of the annual extent of each of the SOM-defined HNLC subregions during the analyzed period (1998-2017). Maximum variation (Max. Variation) is calculated as the difference between the maximum and the minimum extent'.

Figure 1 caption "SOM time-domain analysis at global and regional means" meaning not clear; means of what?

It should read 'and' global and regional mean series of each region. Corrected.

Figure 2 caption "isolines are drawn at 4 mmol m-3 intervals" are they?

No, the reviewer is right. Some of the isolines were removed for clarity. It has been removed.

Figure 4 caption state monthly or annual mean data

We state now that, consistently with the description in M&M section, the spatial patterns were obtained from monthly data.

Figure 6 caption "extent" is still spelled as "extension"

Corrected

Figure 7 - the numbering scheme is unusual. I see the logic of it but it may violate journal standard.

The numbering scheme in Figure 7 has been changed to a more standard notation (a, b….,h)